# ASMIL: Attention-Stabilized Multiple Instance Learning for Whole Slide Imaging

**Linfeng Ye[1], Shayan Mohajer Hamidi[2], Zhixiang Chi[1], Guang Li[3],**
**Mert Pilanci[2], Takahiro Ogawa[3], Miki Haseyama[3], Konstantinos N. Plataniotis[*1]**
[1]University of Toronto, [2]Stanford University, [3]Hokkaido University
[1]`{linfeng.ye,zhixiang.chi}@mail.utoronto.ca`
[1]`kostas@ece.utoronto.ca`
[2]`{smohajer,pilanci}@stanford.edu`
[3]`{guang,ogawa,mhaseyama}@lmd.ist.hokudai.ac.jp`

## Abstract

Attention-based multiple instance learning (MIL) has emerged as a powerful framework for whole slide image (WSI) diagnosis, leveraging attention to aggregate instance-level features into bag-level predictions. Despite this success, we find that such methods exhibit a new failure mode: unstable attention dynamics. Across four representative attention-based MIL methods and two public WSI datasets, we observe that attention distributions oscillate across epochs rather than converging to a consistent pattern, degrading performance. This instability adds to two previously reported challenges: overfitting and over-concentrated attention distribution. To simultaneously overcome these three limitations, we introduce attention-stabilized multiple instance learning (ASMIL), a novel unified framework. ASMIL uses an anchor model to stabilize attention, replaces softmax with a normalized sigmoid function in the anchor to prevent over-concentration, and applies token random dropping to mitigate overfitting. Extensive experiments demonstrate that ASMIL achieves up to a 6.49% F1 score improvement over state-of-the-art methods. Moreover, integrating the anchor model and normalized sigmoid into existing attention-based MIL methods consistently boosts their performance, with F1 score gains up to 10.73%. All code and data are publicly available at `https://github.com/Linfeng-Ye/ASMIL`.

## 1 Introduction

Computational pathology, at the intersection of digital imaging, machine learning, and clinical diagnostics, has transformed modern workflows (Verghese et al., 2023). Advances in whole slide imaging (WSI) now allow glass slides to be digitized into gigapixel images (Bacus, 2001), which are central to cancer diagnosis and treatment planning. WSIs preserve rich spatial context and enable large-scale sharing, but their extreme size and sparsity create major challenges: diagnostically relevant regions often occupy only a tiny fraction of the slide, and exhaustive pixel- or tile-level annotations are infeasible in practice. As a result, most datasets provide only weak slide-level labels, making it critical to design methods that learn effectively under weak supervision.

This weakly supervised setting naturally motivates multiple instance learning (MIL) (Keeler et al., 1990; Dietterich et al., 1997; Maron & Lozano-Pérez, 1998). In MIL, a bag of instances is mapped to a single bag-level label. For WSIs, the image is divided into tiles, each treated as an instance, while only the slide-level label is required. This dramatically reduces annotation costs and makes large-scale WSI datasets more practical for research and clinical use.

Early approaches to MIL-based WSI analysis focused on simple aggregation strategies, such as clustering instance features (Xu et al., 2014) or applying global pooling layers (Kraus et al., 2016). A major breakthrough came with the introduction of attention-based MIL (ABMIL) (Ilse et al., 2018), which provided theoretical guidance for neural network-based MIL algorithms and introduced a

---
[*]Correspondence author.

permutation-invariant attention mechanism to aggregate instance information into bag-level representations. ABMIL established a strong baseline for WSI analysis (Shao et al., 2025) and, importantly, enhanced interpretability through visualized attention scores, which is an essential property for clinical adoption. Building on this foundation, subsequent works have refined ABMIL to further improve performance, scalability, and robustness (Xiong et al., 2021; Shao et al., 2021; Zhang et al., 2022; Tang et al., 2023b; Zhang et al., 2024). In particular, TransMIL replaces independent instance weighting with a transformer encoder that explicitly models inter-instance relations within a bag (Shao et al., 2021). As a result, attention-based MIL has become the de facto choice for WSI subtyping not only because it aggregates instance features but also because its attention maps are used as clinical evidence of model interpretability.

Despite its success, attention-based MIL still suffers from three major problems, which we denote as **(PI)**, **(PII)**, and **(PIII)**, and elaborate on in the sequel.

A critical yet underexplored aspect of MIL-based WSI analysis is the convergence behavior of attention mechanisms during training. The gigapixel scale of WSIs, coupled with weak supervision, high variability, and sparsity, makes it difficult for models to consistently identify informative tiles among thousands of candidates. Our investigation reveals that existing MIL algorithms often fail to converge stably on WSI datasets. To the best of our knowledge, we are the first to identify and systematically analyze **(PI)** *unstable attention dynamics*, where attention distributions for individual WSIs oscillate substantially across epochs instead of converging into consistent patterns. To quantify this phenomenon, we measure the Jensen-Shannon divergence (Cover, 1999) between consecutive attention distributions of the same WSI, as illustrated for TransMIL (Shao et al., 2021) in Figure 1. Additional experiments across methods and datasets are provided in Appendix P. This persistent oscillation results in unstable training and degraded performance, reflected in higher cross-entropy values compared to our proposed method.

Beyond this new limitation identified in our study, prior work has highlighted two additional challenges. One is **(PII)** *over-concentrated attention distribution* (Zhang et al., 2024; Lu et al., 2021), where models allocate excessive importance to only a few tiles, thereby harming generalization and interpretability. The other is **(PIII)** *overfitting* (Zhang et al., 2022; Lin et al., 2023), a common issue in histopathology WSI classification caused by the limited number of available training samples.

In this paper, we aim to simultaneously address the challenges **(PI)**–**(PIII)**. To stabilize attention distribution and the training process, we introduce an *anchor model*, which has the same architecture as the online model's attention module and receives the same input, but is updated via an exponential moving average (EMA) instead of by backpropagation. Acting as a stable reference, the anchor provides smoother and more consistent attention distributions. To transfer this stability, we encourage the online model to mimic the anchor by minimizing the Kullback–Leibler (KL) divergence between their attention distributions. To mitigate over-concentration, which we attribute to the exponential sensitivity of the softmax function, we replace softmax in the anchor branch with a normalized sigmoid function (NSF), as defined in Equation (5). Finally, we propose a simple yet effective token dropout strategy that regularizes the model and reduces overfitting. Together with the anchor model, these components form a unified framework called attention-stabilized multiple instance learning (ASMIL), which improves both the stability and generalization of MIL-based WSI analysis.

In summary, this paper's contributions are as follows:

• We are the first to identify and systematically analyze the problem of *unstable attention dynamics* in attention-based MIL for WSI analysis. This overlooked issue not only limits predictive performance but also undermines interpretability, since fluctuating attention distributions prevent consistent identification of the tissue regions that drive the model's decisions.

• To overcome this instability, we introduce an anchor model that stabilizes attention distribution throughout training. The anchor model is updated using an exponential moving average of the online model, which ensures stable training dynamics and improves both performance and interpretability.

• We show *mathematically* that replacing softmax with an NSF alleviates attention over-concentration. Since applying the NSF to the online model causes vanishing gradients, we apply it to the anchor model instead, ensuring stable and well-distributed attention.

• To mitigate overfitting, we introduce token dropout, which randomly discards a portion of feature tokens during training while retaining all tokens during inference.

• By integrating these innovations, we present attention-stabilized MIL (ASMIL), a novel MIL-based WSI analysis algorithm. Through comprehensive experiments on multiple public WSI datasets, we demonstrate that ASMIL achieves state-of-the-art performance in subtyping and localization tasks.

*Paper Organization.* The remainder of this paper is structured as follows: Section 2 reviews related work on MIL and attention mechanisms in WSI analysis; Section 3 presents the preliminaries and motivation of our approach; Section 4 details the ASMIL framework; Section 5 presents the experimental setup and results; and finally Section 6 concludes the paper with future research directions.

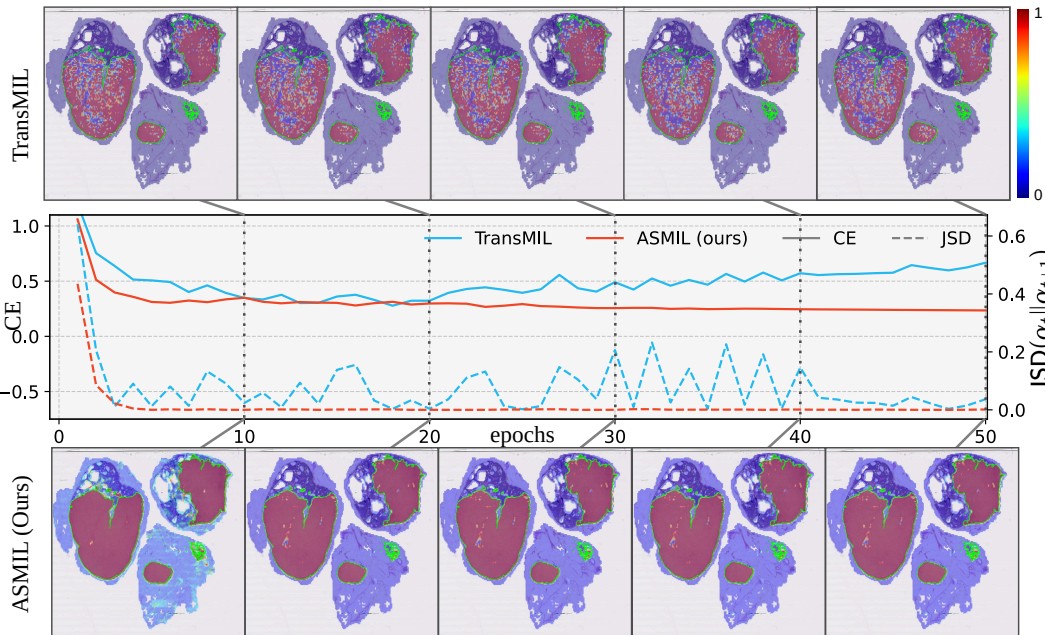

Figure 1: Visualization of attention dynamics on a tumor WSI for TransMIL (Shao et al., 2021) vs. ASMIL (our method). The green contours in the figures indicate the annotated tumor regions. **Top**: TransMIL attention distribution at selected training iterations. **Middle**: Jensen-Shannon divergence (JSD) between attention distributions at successive steps and the cross entropy loss (CE), comparing TransMIL (blue) and ASMIL (red). **Bottom**: Attention distribution from ASMIL over different training iterations. Due to the weakly supervised nature of WSI subtyping datasets, TransMIL's attention patterns never converge during training, further, it focuses on only a subset of cancerous regions. In contrast, our method $(i)$ produces stable attention distributions throughout training and $(ii)$ consistently highlights cancerous regions.

## 2 RELATED WORK

Early weakly supervised approaches in computational pathology leveraged multi-view convolutional neural network ensembles and basic MIL pooling to transition from patch-level labels to slide-level predictions (Das et al., 2017; 2018). As datasets scaled and slide-level supervision became the norm, methods shifted from fixed pooling to attention mechanisms that make aggregation learnable. Building on this trend, attention-based MIL (Ilse et al., 2018) introduced learnable instance weights and generated heatmaps from slide-level labels, achieving breast and colon cancer classification on par with fully supervised methods at scale. Complementary to weighting instances, subsequent work reduced morphological redundancy in tile representations, Song et al. (2024) used a Gaussian mixture model, and sped up inference by skipping irrelevant patches (Dong et al., 2025). Li et al. (2021b) propose DSMIL, a dual-stream MIL framework that selects a critical instance via max-pooling and then applies a trainable non-local, distance-based attention from this instance to all others to form bag embeddings for WSI classification. Subsequent works extend this line of research

by leveraging multi-scale fusion to aggregate information across resolutions (Zhang et al., 2021; Guo et al., 2023; Tran et al., 2025; Buzzard et al., 2024; Li et al., 2019).

Several works further refine training strategies for attention-based MIL. To prevent the attention distribution from collapsing onto a few input patches and to obtain more faithful attention maps, Zhang et al. (2024) stochastically masks the top-$K$ instances, while Zhang et al. (2025b) adds an entropy regularization term that explicitly flattens the attention distribution. In a complementary direction, Fourkioti et al. (2024) introduces neighbor-constrained attention to suppress noise in the feature maps. Because WSI datasets usually contain only a few hundred training samples, many methods focus on mitigating overfitting, for example, by introducing bag splitting to create pseudo-bags (Zhang et al., 2022), designing efficient instance-based classifiers (Qu et al., 2024), and performing hard-negative mining with EMA teachers (Tang et al., 2023b). Lu et al. (2021) introduce clustering-constrained attention multiple-instance learning (CLAM), which replaces max-pooling with class-specific attention pooling and adds instance-level clustering supervision so that weakly supervised slide-level MIL can be both data-efficient and interpretable on WSIs, or using contrastive critical-instance branches (Li et al., 2021a). Recently, Zhu et al. (2025; 2023) systematically studied the effect of random dropping in MIL and proposed to randomly remove the top-$K$ instances with the highest attention weights together with $G \times k$ similar tokens during training, which mitigates overfitting and encourages convergence to flatter regions of the loss landscape, thereby improving generalization. Since our anchor leverages an EMA update, we relate it to EMA/teacher models and provide additional details in Appendix A.

## 3 PRELIMINARIES AND MOTIVATION

### 3.1 NOTATION

Scalars are denoted by non-bold letters (e.g., $a, \beta$), vectors by bold lowercase letters (e.g., $\boldsymbol{a}$), and matrices by bold uppercase letters (e.g., $\boldsymbol{A}$). The $i$-th entry of a vector $\boldsymbol{a}$ is written as $\boldsymbol{a}_i$. A $C$-dimensional probability simplex is denoted by $\Delta^C$. For two distributions $P_1, P_2 \in \Delta^C$, the Kullback–Leibler divergence (KL divergence) is defined as $\mathsf{KL}(P_1 \| P_2) = \sum_{c=1}^{C} P_1[c] \log \frac{P_1[c]}{P_2[c]}$.

### 3.2 MULTIPLE INSTANCE LEARNING WITH ATTENTION

In MIL, supervision is provided only at the bag level. A slide is represented as a bag $X = \{\boldsymbol{x}_i\}_{i=1}^N$ with unknown instance labels. After a pretrained encoder, we obtain instance embeddings $\{\boldsymbol{h}_i\}_{i=1}^N$.

Attention-based MIL assigns a scalar *attention score* to each embedding via a learnable scorer $f_{\boldsymbol{\theta}}$:

$$z_i = f_{\boldsymbol{\theta}}(\boldsymbol{h}_i), \qquad \boldsymbol{z} = (z_1, \dots, z_N) \in \mathbb{R}^N. \tag{1}$$

Scores are normalized into an *attention distribution* on the probability simplex $\Delta^N$ using a softmax:

$$\alpha_i = \frac{\exp(z_i)}{\sum_{j=1}^N \exp(z_j)}, \qquad \sum_{i=1}^N \alpha_i = 1, \qquad \boldsymbol{\alpha} = (\alpha_1, \dots, \alpha_N) \in \Delta^N. \tag{2}$$

The slide-level representation, $\boldsymbol{h}_{\text{bag}} = \sum_{i=1}^N \alpha_i \boldsymbol{h}_i$, is a convex combination of instance features weighted by the attention distribution and is passed to a classifier to produce the bag-level prediction.

### 3.3 MOTIVATION

MIL is effective for WSI analysis, but its weak supervision and small WSI dataset sizes introduce three failure modes: unstable attention dynamics, over-concentrated attention, and overfitting.

● **(PI)** *Unstable attention dynamics.* Under bag-level supervision, we empirically observe that attention distribution oscillates across epochs rather than converging to a consistent pattern. To the best of our knowledge, this phenomenon has not been previously identified or explicitly addressed in the literature. To quantify stability, we measure the Jensen-Shannon divergence (JSD) between consecutive attention distributions for the same WSI. Let $\boldsymbol{\alpha}_t \in \Delta^N$ denote the attention over $N$ tiles at epoch $t$. With $\mathsf{KL}(\cdot \| \cdot)$ denoting the KL divergence and $\bar{\boldsymbol{\alpha}} = \frac{1}{2}(\boldsymbol{\alpha}_t + \boldsymbol{\alpha}_{t+1})$, we define

$$\mathsf{JSD}(\boldsymbol{\alpha}_t \| \boldsymbol{\alpha}_{t+1}) = \tfrac{1}{2}\mathsf{KL}(\boldsymbol{\alpha}_t \| \bar{\boldsymbol{\alpha}}) + \tfrac{1}{2}\mathsf{KL}(\boldsymbol{\alpha}_{t+1} \| \bar{\boldsymbol{\alpha}}). \tag{3}$$

As shown in Figure 1, TransMIL (Shao et al., 2021) exhibits large JSD fluctuations, indicating a lack of stable convergence. Similar behavior appears in other attention-based MIL models; additional results are provided in Appendix P.

• **(PII)** *Over-concentration of attention.* Complementary to instability, prior works report that ABMIL often assigns most mass to a few tiles, which harms generalization and interpretability (Zhang et al., 2024; 2025b). Distinct from previous approaches, we attribute these over-concentrated attention distributions to the exponential nature of the softmax function.

• **(PIII)** *Overfitting.* WSI datasets typically contain only a few slides per class and highly redundant tiles (Zhang et al., 2022). High-capacity neural-network-based MIL models can memorize spurious tile-level patterns, leading to poor out-of-distribution performance. To alleviate this, we introduce a random token drop mechanism specialized for our method.

In the next section, we present our proposed methodology, which simultaneously addresses the three problems **(PI)**, **(PII)**, and **(PIII)**.

## 4 METHODOLOGY

To address the limitations of attention-based MIL, we propose a framework illustrated in Figure 2. Our methodology addresses **(PI)** by stabilizing attention through an anchor model, tackles **(PII)** by replacing softmax with an NSF in the anchor, and mitigates **(PIII)** by token random dropping to regularize training. The next subsections detail each component and the overall objective.

### 4.1 STABILIZING ATTENTION DISTRIBUTIONS VIA AN ANCHOR MODEL

As discussed in Section 3.3, weak supervision in MIL often leads to unstable attention distributions that fluctuate across epochs, preventing convergence. To mitigate this, we introduce an *anchor model* that mirrors the attention block of the online model. The anchor serves as a stable reference by being updated through an EMA of the online model's parameters. Specifically, at training step $t$, the anchor parameters $\boldsymbol{\theta}'_t$ are updated as

$$\boldsymbol{\theta}'_t \leftarrow m\boldsymbol{\theta}'_{t-1} + (1-m)\boldsymbol{\theta}_t, \quad (4)$$

where $\boldsymbol{\theta}_t$ are the online model's parameters and $m \in [0,1)$ is the EMA factor. Both the anchor and online models receive the same inputs, but **only** the online model is updated by backpropagation, the anchor is updated via EMA. The goal is to align the online attention distribution to the anchor distribution, which yields a stabilization loss.

In Appendix C, we show that standard attention-based MIL yields poorly separated bag-level feature clusters during training because attention distributions do not converge reliably. Introducing the an-

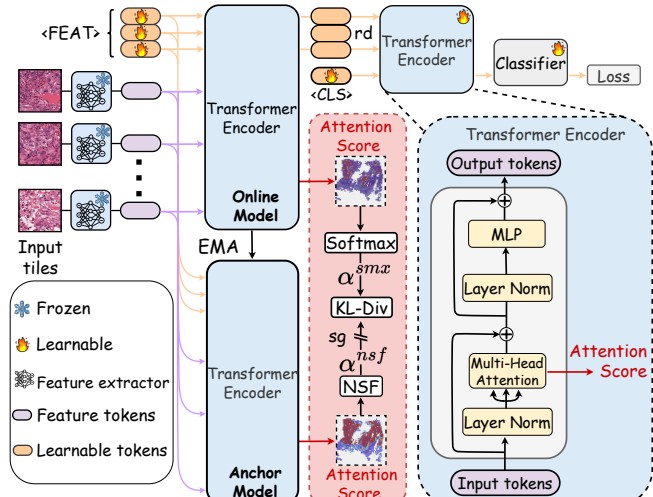

Figure 2: Overview of ASMIL. Each WSI is divided into tiles and embedded into vision tokens using a pretrained encoder. These tokens, along with trainable FEAT tokens, feed into both online and anchor encoders. The anchor encoder's attention scores over the FEAT tokens are transformed into a probability vector using an NSF, while the online encoder applies a softmax. To stabilize training and prevent the online model's attention from becoming overly concentrated, we compute the KL divergence between the two distributions. Gradients are blocked to the anchor encoder using a stop-gradient (sg) operator, and its parameters are updated via EMA from the online encoder. During training, we randomly drop (rd) $N$ FEAT tokens, feed the remaining tokens into a second transformer with a trainable [CLS] token, and train a classifier on its output. 🔥 and ❄ indicate learnable and frozen components, respectively.

chor model stabilizes attention, improves convergence, and produces clearly separated bag-level clusters.

**Remark 1.** *Why an anchor model instead of a single regularizer. Scalar penalties on attention, such as entropy, $\ell_2$, or temperature, are content-agnostic and act only on the current batch. They cannot encode relational structure among instances. An EMA anchor model yields a data-dependent attention distribution conditioned on the bag. Encouraging the online attention to stay close to this target performs functional regularization that captures inter-instance relations and stabilizes training, which a scalar regularizer cannot do.*

The anchor is discarded at inference, adding no extra FLOPs or latency. In the next subsection, we describe how we further improve the anchor's attention using an NSF, which alleviates over-concentration before applying this stabilization loss.

## 4.2 PREVENTING ATTENTION CONCENTRATION WITH NSF IN THE ANCHOR MODEL

In conventional transformer architectures, the softmax function maps self-attention scores $z \in \mathbb{R}^N$ to a probability vector. However, softmax often produces over-concentrated attention, in which a few tokens dominate while the weights of the remaining tokens vanish. Temperature scaling is an incomplete remedy: small temperatures preserve concentration, while large temperatures flatten the distribution so aggressively that weak tokens receive undue weight. We therefore seek a mechanism that equalizes attention among genuinely informative tokens while suppressing weak ones.

We compare softmax with normalized sigmoid function (NSF).[1] For $z = (z_1, \ldots, z_N)$, define

$$\alpha_i^{\text{smx}}(z; T) = \frac{e^{z_i/T}}{\sum_{j=1}^{N} e^{z_j/T}}, \qquad \alpha_i^{\text{nsf}}(z) = \frac{\sigma(z_i)}{\sum_{j=1}^{N} \sigma(z_j)}, \qquad \sigma(t) = \frac{1}{1 + e^{-t}}. \quad (5)$$

For thresholds $\tau > 0$ and bandwidth $\gamma \geq 0$, let $\mathcal{S}(\tau, \gamma, \mathcal{H}, \mathcal{L})$ be the set of score vectors with "high" indices $\mathcal{H}$ satisfying $z_i \in [\tau, \tau + \gamma]$ for $i \in \mathcal{H}$ and "low" indices $\mathcal{L}$ satisfying $z_j \leq -\tau$ for $j \in \mathcal{L}$. Denote $h \triangleq |\mathcal{H}|$ and $\ell \triangleq |\mathcal{L}|$. The following theorem (proof deferred to Appendix E) formalizes the selective flattening property of NSF and shows that softmax cannot match it with a single temperature.

**Theorem 1** (NSF achieves selective flattening; softmax cannot with a single $T$). *Fix $\tau > 0$, $\gamma \geq 0$, and index sets $\mathcal{H}, \mathcal{L}$ with $h \geq 1$, $\ell \geq 1$. For any $z \in \mathcal{S}(\tau, \gamma, \mathcal{H}, \mathcal{L})$:*

*(A) NSF bounds. For any $i, h' \in \mathcal{H}$ and any $j \in \mathcal{L}$,*

$$\frac{\alpha_i^{\text{nsf}}(z)}{\alpha_{h'}^{\text{nsf}}(z)} = \frac{\sigma(z_i)}{\sigma(z_{h'})} \leq \frac{\sigma(\tau + \gamma)}{\sigma(\tau)} = \frac{1 + e^{-\tau}}{1 + e^{-(\tau+\gamma)}} \leq 1 + e^{-\tau}, \quad \alpha_j^{\text{nsf}}(z) \leq \frac{\sigma(-\tau)}{h\,\sigma(\tau)} = \frac{e^{-\tau}}{h}. \quad (6)$$

*Hence, NSF equalizes the high tokens up to a factor $1 + e^{-\tau}$ and suppresses lows to at most $e^{-\tau}/h$. As $\tau \to \infty$ with fixed $\gamma$, ratios among high tokens approach 1 and low-token weights vanish.*

*(B) Softmax incompatibility with one temperature. Suppose we desire suppression and equalization targets $(\varepsilon, \kappa)$ on $\mathcal{S}(\tau, \gamma, \mathcal{H}, \mathcal{L})$:*

$$(\textit{Suppression}) \quad \alpha_j^{\text{smx}}(z; T) \leq \varepsilon \;\; \forall j \in \mathcal{L}, \qquad (\textit{Equalization}) \quad \frac{\max_{i \in \mathcal{H}} \alpha_i^{\text{smx}}(z; T)}{\min_{h' \in \mathcal{H}} \alpha_{h'}^{\text{smx}}(z; T)} \leq \kappa.$$

*Then $T$ must satisfy $T \leq \dfrac{2\tau}{\log\left(\frac{h}{\varepsilon}\right)}$ and $T \geq \dfrac{\gamma}{\log \kappa}$ simultaneously, which is impossible whenever $\dfrac{\gamma}{\log \kappa} > \dfrac{2\tau}{\log\left(\frac{h}{\varepsilon}\right)}$. Thus, no single temperature achieves both targets for all $z \in \mathcal{S}(\tau, \gamma, \mathcal{H}, \mathcal{L})$.*

We further illustrate this effect in Figure 3 by comparing attention maps with softmax and NSF using ABMIL (Ilse et al., 2018) on a cancer slide from the CAMELYON-16 dataset (Ehteshami Bejnordi et al., 2017). Softmax yields a highly concentrated map that obscures broader context, whereas NSF produces a less concentrated attention map that highlights most cancerous regions.

---

[1]We discuss alternatives to NSF, including entmax and softmax with temperature scaling in Appendix F.

A naive option is to apply NSF directly in the online model. In practice, this induces vanishing gradients and degrades performance; see Appendix G. We therefore place NSF in the *anchor* model as a stable prior, guiding the online model without hindering its learning dynamics.

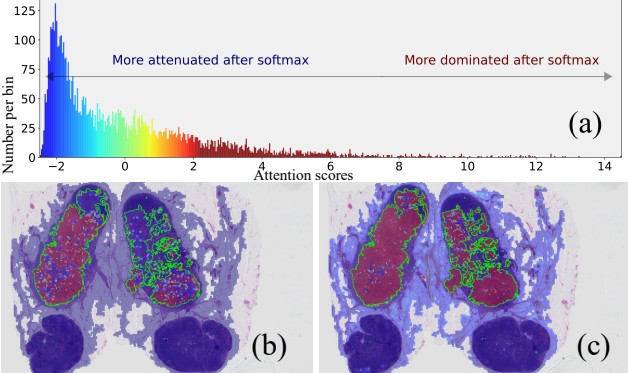

As attention distributions lie on the probability simplex, we use the KL divergence to align the online attention distribution with the NSF-based anchor distribution:

$$\mathcal{L}_{\mathrm{AS}} = \mathsf{KL}(\boldsymbol{\alpha}^{\mathrm{nsf}} \,\|\, \boldsymbol{\alpha}), \quad (7)$$

where $\boldsymbol{\alpha}$ is the online attention (softmax over $\boldsymbol{z}$) and $\boldsymbol{\alpha}^{\mathrm{nsf}}$ is the anchor attention (NSF over the anchor scores). Using $\frac{\partial \alpha_j}{\partial z_i} = \alpha_j(\delta_{ij} - \alpha_i)$ and treating $\boldsymbol{\alpha}^{\mathrm{nsf}}$ as fixed, the gradient with respect to the online attention score $z_i$ is

$$\frac{\partial \mathsf{KL}(\boldsymbol{\alpha}^{\mathrm{nsf}} \,\|\, \boldsymbol{\alpha})}{\partial z_i} = \sum_{j=1}^{N} \alpha_j^{\mathrm{nsf}}(\delta_{ij} - \alpha_i)$$

$$= \alpha_i - \alpha_i^{\mathrm{nsf}}. \quad (8)$$

Figure 3: (a) Distribution of attention scores in ABMIL, which exhibits a long-tailed pattern. (b) Attention distribution obtained with the softmax function and (c) with the NSF. Unlike softmax, the normalized sigmoid suppresses large values in the long tail, yielding a less sparse and more interpretable attention distribution.

Thus, gradient descent moves the online attention toward the anchor distribution, promoting stability and discouraging over-concentration.

**Remark 2.** *The anchor in ASMIL superficially resembles the teacher in MHIM-MIL (Tang et al., 2023b): both are EMA-updated copies of the online model. Their roles, however, differ in two important ways. **(i)** MHIM-MIL uses the teacher to mine hard instances, whereas ASMIL uses the anchor to stabilize attention and prevent over-concentration. **(ii)** MHIM-MIL matches softmax bag-level features, while ASMIL directly matches attention distributions. Appendix I discusses why softmax bag-level matching fails to stabilize attention maps.*

### 4.3 MITIGATING OVERFITTING WITH TOKEN RANDOM DROPPING

To reduce overfitting, we designed a token-level regularizer, specialized for ASMIL, that operates on the trainable tokens used by the online model. Let a WSI $\boldsymbol{x}$ be partitioned into $M$ tiles and embedded by a pretrained encoder into tile tokens $\mathcal{T} = \{\boldsymbol{t}_1, \ldots, \boldsymbol{t}_M\}$. We augment these with $N$ trainable FEAT tokens $\mathcal{P} = \{\boldsymbol{p}_1, \ldots, \boldsymbol{p}_N\}$ and feed the concatenation $[\mathcal{T}; \mathcal{P}]$ into the online encoder. After the online encoder, only the FEAT tokens are retained. Since the number of FEAT tokens is much smaller than the tile tokens (*i.e.*, $N \ll M$), this design acts as information aggregation via token reduction.

During training, we sample an independent Bernoulli mask over FEAT tokens and drop a fraction $B \in [0, 1)$ of them. Denote the kept set by $\mathcal{P}_{\mathrm{keep}}$ with $|\mathcal{P}_{\mathrm{keep}}| = \tilde{N} \sim \mathrm{Binomial}(N, 1 - B)$ and $\mathbb{E}[\tilde{N}] = (1 - B)N$. The remaining tokens, together with a trainable [CLS] token, are passed to a second transformer to produce a bag representation $\boldsymbol{h}_{\mathrm{bag}}$, which is then classified to obtain $\hat{y}$. At inference time, no tokens are dropped $(B = 0)$. Since ASMIL stabilize the attention via aligning the anchor model, which assumes a one-to-one correspondence, as thus general instance dropout method, such as MIL-Dropout Zhu et al. (2025), could not be integrated easily.

This stochastic removal prevents co-adaptation among FEAT tokens and discourages the model from over-relying on a subset of tokens, while preserving image content by keeping all FEAT tokens at inference. Empirically, this acts as an effective regularizer that improves generalization. In Appendix K.4 we study the effect of $B$ and observe a consistent peak in performance around $B \approx 0.5$.

### 4.4 OVERALL TRAINING OBJECTIVE

Based on the discussion thus far, we train with a joint objective that couples standard bag-level classification with attention stabilization:

$$\mathcal{L} = \mathcal{L}_{\mathrm{CE}} + \beta \mathcal{L}_{\mathrm{AS}}, \quad (9)$$

Table 1: The F1 score and AUC of different MIL approaches across three WSI datasets. **Bold** and underlined values denote the best and second-best results, respectively.

| Backbone | Method | CAMELYON-16 | | CAMELYON-17 | | BRACS | |
| | | F1 score ↑ | AUC ↑ | F1 score ↑ | AUC ↑ | F1 score ↑ | AUC ↑ |
|---|---|---|---|---|---|---|---|
| ResNet-18 ImageNet Pretrained | ABMIL ICML 2018 | $0.757_{\pm0.020}$ | $0.790_{\pm0.027}$ | $0.508_{\pm0.032}$ | $0.779_{\pm0.021}$ | $0.523_{\pm0.028}$ | $0.723_{\pm0.035}$ |
| | Clam-SB Nature 2021 | $0.742_{\pm0.024}$ | $0.763_{\pm0.049}$ | $0.504_{\pm0.012}$ | $0.778_{\pm0.024}$ | $0.521_{\pm0.046}$ | $0.750_{\pm0.039}$ |
| | TransMIL NeurIPS 2021 | $0.643_{\pm0.088}$ | $0.706_{\pm0.076}$ | $0.499_{\pm0.082}$ | $0.794_{\pm0.053}$ | $0.444_{\pm0.040}$ | $0.732_{\pm0.043}$ |
| | DSMIL CVPR 2021b | $0.736_{\pm0.025}$ | $0.773_{\pm0.034}$ | $0.473_{\pm0.052}$ | $0.705_{\pm0.022}$ | $0.511_{\pm0.052}$ | $0.751_{\pm0.028}$ |
| | DTFD-MIL CVPR 2022 | $0.758_{\pm0.051}$ | $0.815_{\pm0.063}$ | $0.546_{\pm0.010}$ | $0.735_{\pm0.011}$ | $0.469_{\pm0.016}$ | $0.717_{\pm0.032}$ |
| | IBMIL CVPR 2023 | $0.777_{\pm0.009}$ | $0.799_{\pm0.050}$ | $0.533_{\pm0.015}$ | $0.813_{\pm0.092}$ | $0.510_{\pm0.043}$ | $0.726_{\pm0.034}$ |
| | MHIM-MIL ICCV 2023b | $0.752_{\pm0.034}$ | $0.772_{\pm0.026}$ | $0.56_{\pm0.029}$ | $0.815_{\pm0.019}$ | $0.511_{\pm0.022}$ | $0.775_{\pm0.021}$ |
| | ACMIL ECCV 2024 | $0.798_{\pm0.029}$ | $0.841_{\pm0.030}$ | $0.528_{\pm0.053}$ | $0.789_{\pm0.046}$ | $0.552_{\pm0.048}$ | $0.754_{\pm0.008}$ |
| | CAMIL ICLR 2024 | $0.778_{\pm0.011}$ | $0.812_{\pm0.017}$ | $0.503_{\pm0.007}$ | $0.806_{\pm0.006}$ | $0.569_{\pm0.007}$ | $\underline{0.787}_{\pm0.011}$ |
| | AEM MICCAI 2025b | $\underline{0.804}_{\pm0.022}$ | $\underline{0.859}_{\pm0.031}$ | $0.525_{\pm0.043}$ | $0.828_{\pm0.054}$ | $0.554_{\pm0.004}$ | $0.764_{\pm0.008}$ |
| | HDMIL CVPR 2025 | $0.790_{\pm0.023}$ | $0.856_{\pm0.027}$ | $\underline{0.557}_{\pm0.007}$ | $\mathbf{0.853}_{\pm0.013}$ | $\underline{0.578}_{\pm0.012}$ | $0.761_{\pm0.011}$ |
| | ASMIL (Ours) | $\mathbf{0.814}_{\pm0.052}$ | $\mathbf{0.870}_{\pm0.064}$ | $\mathbf{0.564}_{\pm0.020}$ | $\underline{0.851}_{\pm0.061}$ | $\mathbf{0.601}_{\pm0.072}$ | $\mathbf{0.810}_{\pm0.054}$ |
| ViT-S SSL pretrained | ABMIL ICML 2018 | $0.914_{\pm0.031}$ | $0.945_{\pm0.027}$ | $0.522_{\pm0.050}$ | $0.853_{\pm0.016}$ | $0.680_{\pm0.051}$ | $0.866_{\pm0.029}$ |
| | Clam-SB Nature 2021 | $0.925_{\pm0.085}$ | $0.969_{\pm0.024}$ | $0.523_{\pm0.020}$ | $0.846_{\pm0.020}$ | $0.631_{\pm0.034}$ | $0.863_{\pm0.005}$ |
| | TransMIL NeurIPS 2021 | $0.922_{\pm0.019}$ | $0.943_{\pm0.009}$ | $0.554_{\pm0.048}$ | $0.792_{\pm0.029}$ | $0.631_{\pm0.030}$ | $0.841_{\pm0.006}$ |
| | DSMIL CVPR 2021b | $0.943_{\pm0.007}$ | $0.966_{\pm0.009}$ | $0.532_{\pm0.064}$ | $0.804_{\pm0.032}$ | $0.577_{\pm0.028}$ | $0.816_{\pm0.028}$ |
| | DTFD-MIL CVPR 2022 | $0.948_{\pm0.007}$ | $\underline{0.980}_{\pm0.011}$ | $0.627_{\pm0.015}$ | $0.866_{\pm0.012}$ | $0.612_{\pm0.080}$ | $0.870_{\pm0.022}$ |
| | IBMIL CVPR 2023 | $0.912_{\pm0.034}$ | $0.954_{\pm0.022}$ | $0.557_{\pm0.064}$ | $0.850_{\pm0.024}$ | $0.645_{\pm0.041}$ | $0.871_{\pm0.014}$ |
| | MHIM-MIL ICCV 2023b | $0.932_{\pm0.024}$ | $0.970_{\pm0.037}$ | $0.541_{\pm0.022}$ | $0.845_{\pm0.026}$ | $0.625_{\pm0.060}$ | $0.865_{\pm0.017}$ |
| | ACMIL ECCV 2024 | $0.954_{\pm0.012}$ | $0.974_{\pm0.012}$ | $0.562_{\pm0.050}$ | $0.863_{\pm0.004}$ | $0.722_{\pm0.030}$ | $0.888_{\pm0.010}$ |
| | CAMIL ICLR 2024 | $0.930_{\pm0.009}$ | $0.963_{\pm0.011}$ | $0.633_{\pm0.052}$ | $0.886_{\pm0.034}$ | $0.709_{\pm0.011}$ | $0.836_{\pm0.014}$ |
| | AEM MICCAI 2025b | $0.947_{\pm0.003}$ | $0.974_{\pm0.007}$ | $\underline{0.647}_{\pm0.007}$ | $\underline{0.887}_{\pm0.013}$ | $\underline{0.742}_{\pm0.030}$ | $\underline{0.905}_{\pm0.010}$ |
| | HDMIL CVPR 2025 | $\underline{0.958}_{\pm0.013}$ | $0.976_{\pm0.017}$ | $0.571_{\pm0.012}$ | $0.796_{\pm0.022}$ | $0.717_{\pm0.033}$ | $0.874_{\pm0.010}$ |
| | ASMIL (Ours) | $\mathbf{0.965}_{\pm0.020}$ | $\mathbf{0.985}_{\pm0.017}$ | $\mathbf{0.689}_{\pm0.005}$ | $\mathbf{0.898}_{\pm0.010}$ | $\mathbf{0.781}_{\pm0.042}$ | $\mathbf{0.914}_{\pm0.014}$ |

where the coefficient $\beta > 0$ balances the stabilization and classification objectives. In practice, to calculate $\mathcal{L}_{AS}$, $\alpha$ is computed by a softmax over the online scores, $\alpha^{nsf}$ is computed by applying the NSF to the anchor scores, and the anchor model is treated as *stop-gradient* while its parameters are updated via EMA. The KL divergence is taken over the attention distributions on the FEAT token set used for aggregation. This objective discourages attention concentration through $\mathcal{L}_{AS}$ and preserves task performance through $\mathcal{L}_{CE}$. ASMIL can be easily applied to other tasks, including survival prediction by replacing the objective function and the classification head accordingly. During training, the online model is updated by gradient descent

$$\theta_{t+1} = \theta_t - \eta \nabla_\theta \mathcal{L}, \tag{10}$$

where $\eta$ is the learning rate. $\mathcal{L}$ is computed as in Equation (9). The anchor model is then updated according to Equation (4). The gradient is only used to update the online model, while the anchor model influences learning through Equation (7). At inference time, ASMIL uses only the online model and discards the anchor model; therefore, the anchor does not increase the computational budget at inference.

## 5 EXPERIMENTS

To demonstrate the effectiveness of ASMIL, we evaluate it on three well-known public WSI subtyping datasets: ($i$) CAMELYON-16 (Ehteshami Bejnordi et al., 2017), ($ii$) CAMELYON-17 (Bándi et al., 2019), and ($iii$) BRACS (Brancati et al., 2022). Details of the data splits, preprocessing, training setup, and baselines are provided in the Appendix B. We further evaluate ASMIL on survival prediction and non-WSI datasets in Appendix N and Appendix O, respectively.

### 5.1 SUBTYPING PERFORMANCE

We compare ASMIL against eleven attention-based MIL baselines that are designed for WSIs: CLAM-SB (Lu et al., 2021), TransMIL (Shao et al., 2021), DSMIL (Li et al., 2021b), DTFD-MIL (Zhang et al., 2022), IBMIL (Lin et al., 2023), MHIM-MIL (Tang et al., 2023b), ABMIL (Ilse et al., 2018), ACMIL (Zhang et al., 2024), CAMIL (Fourkioti et al., 2024), AEM (Zhang et al., 2025b) and HDMIL (Dong et al., 2025). Because WSI datasets are class-imbalanced, we report the F1 score and area under the ROC curve (AUC) for each dataset in Table 1[2].

---

[2] See Appendix D for details on metric computation and interpretation.

Table 2: Applying anchor model and NSF to other attention-based MIL methods.

| Dataset | | | CAMELYON-16 | | CAMELYON-17 | | BRACS | |
|---|---|---|---|---|---|---|---|---|
| Method | Anchor | NSF | F1 score ↑ | AUC ↑ | F1 score ↑ | AUC ↑ | F1 score ↑ | AUC ↑ |
| ABMIL ICML 2018 | ✗ | ✗ | $0.914_{\pm0.031}$ | $0.945_{\pm0.027}$ | $0.522_{\pm0.050}$ | $0.853_{\pm0.016}$ | $0.680_{\pm0.051}$ | $0.866_{\pm0.029}$ |
| | ✓ | ✗ | $0.951_{\pm0.015}$ +0.037 | $0.963_{\pm0.008}$ +0.018 | $0.573_{\pm0.011}$ +0.051 | $0.871_{\pm0.010}$ +0.018 | $0.751_{\pm0.013}$ +0.071 | $0.877_{\pm0.007}$ +0.011 |
| | ✓ | ✓ | $0.953_{\pm0.009}$ +0.039 | $0.967_{\pm0.006}$ +0.022 | $0.574_{\pm0.010}$ +0.052 | $0.883_{\pm0.014}$ +0.030 | $0.753_{\pm0.009}$ +0.073 | $0.887_{\pm0.014}$ +0.021 |
| CLAM-SB Nature 2021 | ✗ | ✗ | $0.925_{\pm0.085}$ | $0.969_{\pm0.024}$ | $0.523_{\pm0.020}$ | $0.846_{\pm0.020}$ | $0.631_{\pm0.034}$ | $0.863_{\pm0.005}$ |
| | ✓ | ✗ | $0.937_{\pm0.004}$ +0.012 | $0.979_{\pm0.015}$ +0.010 | $0.547_{\pm0.006}$ +0.024 | $0.887_{\pm0.0014}$ +0.041 | $0.678_{\pm0.018}$ +0.047 | $0.866_{\pm0.007}$ +0.003 |
| | ✓ | ✓ | $0.948_{\pm0.014}$ +0.023 | $0.981_{\pm0.021}$ +0.012 | $0.550_{\pm0.006}$ +0.027 | $0.886_{\pm0.0015}$ +0.040 | $0.679_{\pm0.013}$ +0.048 | $0.887_{\pm0.002}$ +0.024 |
| TransMIL NeurIPS 2021 | ✗ | ✗ | $0.922_{\pm0.019}$ | $0.943_{\pm0.009}$ | $0.554_{\pm0.048}$ | $0.792_{\pm0.029}$ | $0.631_{\pm0.030}$ | $0.841_{\pm0.006}$ |
| | ✓ | ✗ | $0.931_{\pm0.001}$ +0.009 | $0.947_{\pm0.008}$ +0.004 | $0.577_{\pm0.006}$ +0.023 | $0.824_{\pm0.012}$ +0.032 | $0.647_{\pm0.024}$ +0.016 | $0.853_{\pm0.021}$ +0.012 |
| | ✓ | ✓ | $0.933_{\pm0.023}$ +0.011 | $0.954_{\pm0.021}$ +0.011 | $0.580_{\pm0.008}$ +0.026 | $0.829_{\pm0.010}$ +0.037 | $0.672_{\pm0.024}$ +0.041 | $0.883_{\pm0.041}$ +0.045 |
| DSMIL CVPR 2021b | ✗ | ✗ | $0.943_{\pm0.007}$ | $0.966_{\pm0.009}$ | $0.532_{\pm0.064}$ | $0.804_{\pm0.032}$ | $0.577_{\pm0.028}$ | $0.816_{\pm0.028}$ |
| | ✓ | ✗ | $0.943_{\pm0.001}$ ±0.000 | $0.974_{\pm0.007}$ +0.008 | $0.544_{\pm0.038}$ +0.012 | $0.819_{\pm0.031}$ +0.015 | $0.609_{\pm0.012}$ +0.032 | $0.837_{\pm0.013}$ +0.021 |
| | ✓ | ✓ | $0.942_{\pm0.026}$ −0.001 | $0.985_{\pm0.022}$ +0.019 | $0.559_{\pm0.028}$ +0.027 | $0.823_{\pm0.019}$ +0.019 | $0.612_{\pm0.031}$ +0.035 | $0.849_{\pm0.042}$ +0.033 |

Overall, ASMIL demonstrates superior performance, achieving state-of-the-art performance on all datasets when paired with an in-domain ViT-SSL backbone, and remains competitive with the best baseline on ImageNet-pretrained ResNet-18 features. On the BRACS dataset, our method attains an F1 score of 0.781 and an AUC of 0.914, exceeding the previous best results by 3.9 and 0.9 percentage points, respectively. This shows its effectiveness in capturing subtle histopathological features in heterogeneous subtyping tasks.

For CAMELYON-16 and CAMELYON-17 datasets with sparse tumor regions, where malignant tissue may occupy as little as 5% of a slide (Cheng et al., 2021),the advantages are even more pronounced. on CAMELYON-16, we observe a 3.3% increase in F1 score and a 1.6% uplift in AUC compared to the strongest baseline; similarly, on CAMELYON-17, ASMIL improves the F1 score by 6.49%, which highlights ASMIL's efficacy under an ill-posed, weakly supervised task. We compare the computational cost of ASMIL with that of other benchmarks in Appendix M.1.

## 5.2 INTEGRATING THE ANCHOR MODEL AND NSF WITH OTHER MIL METHODS

We regard the anchor model as a general plug-in module for attention-based MIL in WSI analysis. Accordingly, for each baseline we evaluate two variants while keeping all other components and hyperparameters fixed: ($i$) **+Anchor** (EMA-updated anchor with attention matching), and ($ii$) **+Anchor+NSF** (anchor updated by EMA and using NSF). The results are summarized in Table 2. As shown, adding the anchor model and the NSF consistently improves performance, with F1 score gains up to 10.73% (for ABMIL on BRACS), except when adding the anchor to DSMIL on the CAMELYON-16 dataset, where the F1 score decreases by 0.001 relative to the original model. The additional computational cost introduced by the anchor model is reported in Appendix M.2.

## 5.3 LOCALIZATION

We evaluate tumor localization on CAMELYON-16 both qualitatively and quantitatively. Qualitative heatmaps are shown in Figure 4. Compared with baseline methods, ASMIL consistently highlights all cancerous regions. We attribute these gains to reduced over-concentration by the NSF in the anchor model, which yields more faithful attention distributions.

Table 3: Component-wise ablation of ASMIL on BRACS. We evaluate the contribution of the anchor model, NSF, and random drop (rd).

| Anchor | NSF | rd | F1 score ↑ | AUC ↑ |
|---|---|---|---|---|
| ✓ | ✓ | ✓ | $\mathbf{0.781_{\pm0.042}}$ | $\mathbf{0.914_{\pm0.014}}$ |
| ✓ | ✓ | ✗ | $0.765_{\pm0.030}$ | $0.903_{\pm0.018}$ |
| ✓ | ✗ | ✓ | $0.759_{\pm0.028}$ | $0.895_{\pm0.012}$ |
| ✓ | ✗ | ✗ | $0.747_{\pm0.026}$ | $0.887_{\pm0.015}$ |
| ✗ | ✗ | ✓ | $0.728_{\pm0.019}$ | $0.868_{\pm0.010}$ |
| ✗ | ✗ | ✗ | $0.712_{\pm0.020}$ | $0.860_{\pm0.012}$ |

Following the official CAMELYON-16 and Fourkioti et al. (2024), we report lesion-level Free-Response ROC (FROC) (Miller, 1969; Bunch, 1978) the Dice coefficient on cancerous slides, and tile-level specificity on normal slides. To obtain the predicted masks, we use scaled attention distributions for CLAM (Lu et al., 2021), TransMIL (Shao et al., 2021), DSMIL (Li et al., 2021b), and CAMIL (Fourkioti et al., 2024); tile-level logits for DTFD-MIL (Zhang et al., 2022); and for ASMIL, the per-tile average of FEAT-token attentions. Quantitative results for FROC, Dice, and specificity, as well as additional attention-map visualizations, are provided in Appendix L.

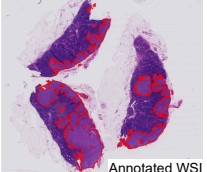 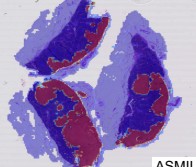 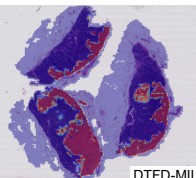 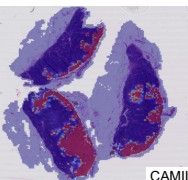 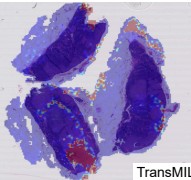

Figure 4: Visual comparison of attention maps on the CAMELYON-16 dataset. The left column shows the original WSI with ground-truth tumor annotations outlined in red; the remaining columns present attention maps for ASMIL (ours), DTFD-MIL, CAMIL, and TransMIL (left to right).

### 5.4 ABLATION STUDY

Lastly, we evaluate the effect of the anchor model, NSF, and random drop (rd) by enabling or disabling them in all combinations. As shown in Table 3, the full model (all three enabled) achieves the best F1 score and AUC. Removing any component degrades performance, with the anchor model having the largest impact. Without all three, the model drops to the lowest scores, confirming that each component contributes to the overall effectiveness of ASMIL. Additional ablations on the loss weight $\beta$, the number of trainable FEAT tokens, the EMA factor $m$, the anchor update frequency, and the random drop rate are reported in Appendix K.

## 6 CONCLUSION

In this work, we identified a previously overlooked failure mode in attention-based MIL for WSI: unstable attention dynamics that hinder convergence. We proposed ASMIL, which stabilizes training via an anchor model, prevents over-concentration by using a normalized sigmoid in the anchor, and mitigates overfitting with token dropout. Across multiple WSI benchmarks, ASMIL improves classification performance and state-of-the-art localization performance. These results underscore the importance of jointly controlling attention stability, concentration, and overfitting in weakly supervised WSI analysis. We anticipate that the proposed anchor model and normalized sigmoid function will serve as building blocks for future MIL-based WSI analysis algorithms, ultimately facilitating more accurate and interpretable analysis of gigapixel pathology images. Due to space constraints, we defer the discussion of future work and limitations to Appendix Q.

## ETHICS STATEMENT

All WSI datasets used in this work are publicly available and were obtained from open-access websites. The usage of these datasets strictly follows the terms and conditions set by the dataset providers and adheres to established academic and research community standards. No personally identifiable information or sensitive patient data is involved.

## REPRODUCIBILITY STATEMENT

We have taken steps to ensure our results are reproducible. All model and algorithmic details, training procedures, hyperparameters, evaluation protocols, and metrics are specified in the main text. The appendix provides complete proofs, implementation notes, ablations, and additional qualitative results. An anonymized GitHub repository contains the source code and configuration files, and pre-trained checkpoints. All datasets used in our experiments are publicly available; download links, data splits, and preprocessing steps are documented in the repository and referenced in the appendix.

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

## A  RELATED WORK ON EMA MODELS AND ANCHORING STRATEGIES

EMA-based target networks are central in self-supervised representation learning. Mean Teacher Tarvainen & Valpola (2017) maintains an EMA of student parameters and enforces prediction consistency with this temporal ensemble under limited supervision. BYOL (Grill et al., 2020; Wu et al., 2023) uses an EMA-updated target network to provide representation targets for an online network with an additional predictor, and avoids collapse through architectural asymmetry and the EMA update instead of negatives. DINO-style methods (Caron et al., 2021; Oquab et al., 2023) adapt EMA self-distillation to Vision Transformers, where an EMA teacher produces soft probability targets on multi-crop views; centering, sharpening, and a momentum schedule control the stability–adaptation trade-off of these targets.

DINOv3 (Siméoni et al., 2025) revisits EMA teachers for dense prediction and studies how dense features drift or collapse under long training. It introduces *Gram anchoring*, which aligns Gram matrices of patch–patch similarities between a student and its EMA teacher so that dense features remain close to a temporally smoothed reference. The EMA momentum and the strength of this anchoring loss jointly determine how strongly dense features are tied to the teacher versus how quickly they adapt.

ASMIL also maintains an EMA-updated copy of the model, but uses it in a different regime and on a different target. Training is fully supervised at the bag level, and the EMA branch does not supply pseudo-labels or representation targets. Instead, it defines a temporally smoothed *attention distribution* over tiles, and the anchor enters the loss only through the KL term in Eq. equation 7, while the bag-level cross-entropy in Eq. equation 9 provides all semantic supervision. The shared encoder and classifier parameters are optimized by standard backpropagation; the EMA update acts purely as a temporal regularizer on attention, in contrast to BYOL/DINO, which anchors global embeddings, and DINOv3, which anchors patch–patch similarity structure.

The same EMA hyperparameters induce an analogous stability–adaptation trade-off but at the level of attention rather than features. The EMA momentum in ASMIL sets how rapidly the anchor follows the online model, and the weight $\beta$ on the KL term controls how strongly attention is pulled toward the temporally smoothed reference. Unlike BYOL and DINO/DINOv3, where EMA model is designed to avoid global representation collapse, ASMIL uses EMA anchoring to reduce unstable and over-concentrated attention patterns observed under purely online MIL training, while the supervised objective already discourages trivial constant-attention solutions.

# B EXPERIMENTAL DETAILS

We train all models for 50 epochs with a batch size of 1, using Adam (weight decay $10^{-4}$) and a cosine learning rate schedule with an initial learning rate of $10^{-4}$. All reported results are averaged over five random seeds.

## B.1 WSI PRE-PROCESSING

For all datasets, we used the publicly available CLAM WSI preprocessing toolbox (Lu et al., 2021) to segment tissue regions and divide each slide into non-overlapping $256 \times 256$ patches at $20 \times$ magnification. Tissue segmentation was performed automatically using Otsu's thresholding. To reduce computational overhead and leverage previously learned representations, we adopted a ResNet-18 model (He et al., 2016) pretrained on ImageNet (Russakovsky et al., 2015) and an open-source self-supervised ViT-small model (Kang et al., 2023) as feature extractors[3]. The ViT-small model was pretrained on 36,666 whole slide images from The Cancer Genome Atlas (TCGA) and the internally collected TULIP dataset. For consistency and fairness in the subtyping task, we used the same feature extractors across all baseline methods.

For the localization experiments, following Tourniaire et al. (2023), we used a ResNet-18 backbone pretrained with SimCLR (Chen et al., 2020)[4]. This feature extractor maps each tile to a 1024-dimensional feature vector.

## B.2 DATASETS

CAMELYON-16 (Ehteshami Bejnordi et al., 2017) is a widely used publicly available WSI dataset designed for lymph node metastasis detection. It contains 270 training and 129 test slides collected from two medical centers, with detailed pixel-level annotations provided by expert pathologists. Notably, some slides include only partial annotations, making the dataset particularly challenging due to the presence of small or sparse metastatic regions. CAMELYON-16 has become a standard benchmark for evaluating weakly supervised and fully supervised algorithms in computational pathology.

CAMELYON-17 (Bándi et al., 2019) extends the scope of CAMELYON-16 by including a total of 1,000 WSIs from five medical centers, making it a more diverse and clinically representative dataset. Among these, 500 slides are publicly available and come with slide-level labels, while the remaining 500 are held out for challenge-based evaluations. The inclusion of data from multiple institutions introduces significant variability in staining and scanning conditions, making CAMELYON-17 a suitable benchmark for testing the generalization performance of WSI-based models.

The BRACS dataset (Brancati et al., 2022) is a large-scale WSI dataset curated for the task of breast cancer subtype classification. It comprises 547 WSIs collected from several medical institutions and annotated by expert pathologists into clinically relevant categories: benign tumors, atypical tumors, and malignant tumors. These labels reflect the progression of breast lesions and are critical for diagnostic decision-making and treatment planning. BRACS captures a wide range of histological appearances and staining variations, making it a valuable resource for developing and benchmarking MIL and weakly supervised classification models in real-world clinical settings.

## B.3 DATA SPLITS

Following Zhang et al. (2025b; 2024), we partition the datasets as follows. For CAMELYON-16, the WSIs are divided into training, validation, and test sets. The 270 WSIs from Hospital 1 are split, five times, into training (90%) and validation (10%) subsets; the 130 WSIs from Hospital 2 are used as a test set. The official test set of 129 WSIs is used for final evaluation. For CAMELYON-17, we use 500 WSIs in total: 300 WSIs from three hospitals for training/validation (90%, 10%) and 200 WSIs from two other hospitals for testing to assess out-of-distribution (OOD) performance. For BRACS, we follow the official split: 395 slides for training, 65 for validation, and 87 for testing.

---

[3]The checkpoint is available at `https://github.com/lunit-io/benchmark-ssl-pathology`.

[4]The checkpoint is available at `https://github.com/binli123/dsmil-wsi`.

The task is a three-class WSI classification—benign tumor, atypical tumor, and malignant tumor. All results are averaged over five random seeds, and we report the mean performance on the official competition test set.

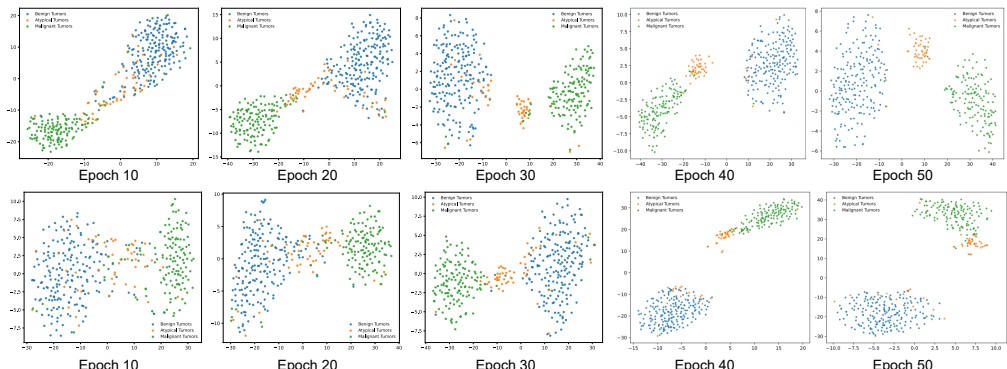

Figure 5: T-SNE embeddings of ASMIL bag-level features on the BRACS training set across training epochs. **Top:** with the anchor model; **Bottom:** without the anchor model.

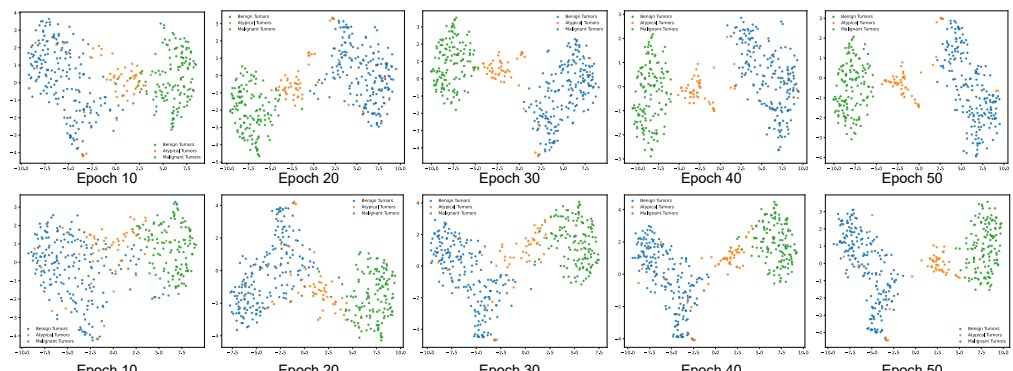

Figure 6: T-SNE embeddings of TransMIL bag-level features on the BRACS training set across training epochs. **Top:** with the anchor model; **Bottom:** without the anchor model.

## C T-SNE VISUALIZATION OF BAG-LEVEL FEATURES

To assess how the anchor model stabilizes attention during training, we visualize the bag-level representations learned by ASMIL using t-SNE Maaten & Hinton (2008); see Figure 5. Compared to ASMIL without the anchor, the model with an anchor forms more distinct clusters and exhibits clearer inter-class boundaries across training epochs, indicating faster convergence and more discriminative features. We observe a similar trend for TransMIL in Figure 6.

## D MACRO AUC AND MACRO F1 SCORE UNDER CLASS IMBALANCE

Since all datasets considered in this work are class-imbalanced, we report *macro-averaged* variants of the area under the ROC curve (AUC) and the F1 score as our primary summary metrics. Macro-averaging assigns equal weight to each class and therefore prevents majority classes from dominating the overall score.

**Setup.** Let $\mathcal{Y} = \{1, \ldots, K\}$ denote the set of classes. For a sample $x$ with true label $y \in \mathcal{Y}$, let $s_k(x) \in \mathbb{R}$ be the model score for class $k$. Define one-vs-rest binary indicators $y_k = \mathbb{1}[y = k]$ for each class $k$, and the corresponding confusion-matrix counts $(\mathrm{TP}_k, \mathrm{FP}_k, \mathrm{FN}_k, \mathrm{TN}_k)$ computed by treating class $k$ as "positive" and all others as "negative".

### D.1 MACRO-$F1$

For class $k$, precision and recall are

$$\text{Precision}_k = \frac{\text{TP}_k}{\text{TP}_k + \text{FP}_k}, \qquad \text{Recall}_k = \frac{\text{TP}_k}{\text{TP}_k + \text{FN}_k}. \tag{11}$$

The per-class $F1$ is the harmonic mean of precision and recall:

$$F1_k = \frac{2\,\text{Precision}_k\,\text{Recall}_k}{\text{Precision}_k + \text{Recall}_k}. \tag{12}$$

The *macro-$F1$* averages the per-class values uniformly:

$$\text{Macro-}F1 = \frac{1}{K}\sum_{k=1}^{K} F_{1,k}. \tag{13}$$

As a thresholded, decision-level metric, $F1_k$ (and thus macro-$F1$) depends on the classification threshold applied to scores $s_k(x)$. We use a threshold of $0.5$ for all experiments. The same definition applies to multilabel settings by averaging over labels.

### D.2 MACRO-AUC (ROC)

For class $k$, the ROC curve plots the true positive rate against the false positive rate as the threshold on $s_k(x)$ varies:

$$\text{TPR}_k = \frac{\text{TP}_k}{\text{TP}_k + \text{FN}_k}, \qquad \text{FPR}_k = \frac{\text{FP}_k}{\text{FP}_k + \text{TN}_k}. \tag{14}$$

The per-class AUC, $\text{AUC}_k \in [0,1]$, is the area under this curve; equivalently, it is the probability that a randomly chosen positive example (for class $k$) receives a higher score than a randomly chosen negative example. The *macro-AUC* is the uniform average across classes:

$$\text{Macro-AUC} = \frac{1}{K}\sum_{k=1}^{K} \text{AUC}_k. \tag{15}$$

Unlike $F1$, AUC is threshold-agnostic and measures the ranking quality of scores.

## E   PROOF OF THEOREM 1

*Proof.* We proceed in two parts.

**Part A: NSF bounds.** Let $s_i = \sigma(z_i)$ and $S = \sum_{j=1}^{N} \sigma(z_j)$, so $\alpha_i^{\text{nsf}} = s_i/S$.

*Equalization among highs.* For $i, h' \in \mathcal{H}$,

$$\frac{\alpha_i^{\text{nsf}}}{\alpha_{h'}^{\text{nsf}}} = \frac{s_i}{s_{h'}} = \frac{\sigma(z_i)}{\sigma(z_{h'})}. \tag{16}$$

Since $\sigma$ is strictly increasing and $z_i, z_{h'} \in [\tau, \tau + \gamma]$,

$$\frac{\sigma(z_i)}{\sigma(z_{h'})} \leq \frac{\sigma(\tau + \gamma)}{\sigma(\tau)} = \frac{1 + e^{-\tau}}{1 + e^{-(\tau+\gamma)}} \leq 1 + e^{-\tau}. \tag{17}$$

*Suppression of lows.* For any $j \in \mathcal{L}$ we have $z_j \leq -\tau$. Using monotonicity and the identity

$$\sigma(-t) = e^{-t}\,\sigma(t) \qquad \text{for all } t \in \mathbb{R}, \tag{18}$$

we get $\sigma(z_j) \leq \sigma(-\tau) = e^{-\tau}\sigma(\tau)$. Meanwhile

$$S = \sum_{i=1}^{N} \sigma(z_i) \geq \sum_{i \in \mathcal{H}} \sigma(z_i) \geq h\,\sigma(\tau), \tag{19}$$

since $z_i \geq \tau$ for $i \in \mathcal{H}$. Hence

$$\alpha_j^{\text{nsf}} = \frac{\sigma(z_j)}{S} \leq \frac{e^{-\tau}\sigma(\tau)}{h\,\sigma(\tau)} = \frac{e^{-\tau}}{h}. \tag{20}$$

For completeness, equation 18 follows from $\sigma(-t) = \frac{1}{1+e^t} = \frac{e^{-t}}{1+e^{-t}} = e^{-t}\sigma(t)$.

**Part B: Softmax temperature constraints.** Fix $T > 0$ and $\boldsymbol{z} \in \mathcal{S}(\tau, \gamma, \mathcal{H}, \mathcal{L})$.

*Equalization among highs.* For any $i, h' \in \mathcal{H}$,

$$\frac{\alpha_i^{\text{smx}}}{\alpha_{h'}^{\text{smx}}} = \frac{e^{z_i/T}}{e^{z_{h'}/T}} = e^{(z_i - z_{h'})/T}. \tag{21}$$

Over $\mathcal{S}(\tau, \gamma, \mathcal{H}, \mathcal{L})$, the worst high to high ratio occurs at $z_i = \tau + \gamma$ and $z_{h'} = \tau$, so

$$\frac{\max_{i \in \mathcal{H}} \alpha_i^{\text{smx}}}{\min_{h' \in \mathcal{H}} \alpha_{h'}^{\text{smx}}} \geq e^{\gamma/T}. \tag{22}$$

Therefore, the uniform bound $\frac{\max_{i \in \mathcal{H}} \alpha_i^{\text{smx}}}{\min_{h' \in \mathcal{H}} \alpha_{h'}^{\text{smx}}} \leq \kappa$ for all $\boldsymbol{z} \in \mathcal{S}(\tau, \gamma, \mathcal{H}, \mathcal{L})$ implies

$$T \geq \frac{\gamma}{\log \kappa}. \tag{23}$$

*Suppression of lows.* Fix $j \in \mathcal{L}$. For a given $T$, the quantity $\alpha_j^{\text{smx}}(\boldsymbol{z}; T)$ is maximized over $\mathcal{S}(\tau, \gamma, \mathcal{H}, \mathcal{L})$ by taking $z_j = -\tau$, $z_i = \tau\ \forall i \in \mathcal{H}$, $z_k \to -\infty$ for $k \notin \mathcal{H} \cup \{j\}$, which minimizes the denominator subject to the constraints. Thus

$$\sup_{\boldsymbol{z} \in \mathcal{S}(\tau, \gamma, \mathcal{H}, \mathcal{L})} \alpha_j^{\text{smx}}(\boldsymbol{z}; T) = \frac{e^{-\tau/T}}{h\,e^{\tau/T} + e^{-\tau/T}} = \frac{1}{h\,e^{2\tau/T} + 1}. \tag{24}$$

Consequently, the uniform suppression requirement $\alpha_j^{\text{smx}}(\boldsymbol{z}; T) \leq \varepsilon$ for all $\boldsymbol{z} \in \mathcal{S}(\tau, \gamma, \mathcal{H}, \mathcal{L})$ forces

$$\frac{1}{h\,e^{2\tau/T} + 1} \leq \varepsilon \iff h\,e^{2\tau/T} \geq \frac{1}{\varepsilon} - 1 \iff T \leq \frac{2\tau}{\log\left(\frac{1}{\varepsilon} - 1\right) - \log h}. \tag{25}$$

Combining equation 23 and equation 25 yields the simultaneous constraints $T \leq \frac{2\tau}{\log\left(\frac{1}{\varepsilon} - 1\right) - \log h}$, $T \geq \frac{\gamma}{\log \kappa}$. If

$$\frac{\gamma}{\log \kappa} > \frac{2\tau}{\log\left(\frac{1}{\varepsilon} - 1\right) - \log h}, \tag{26}$$

no $T$ can satisfy both.

*Instantiating NSF targets.* Set $\varepsilon = \varepsilon_{\text{nsf}} = e^{-\tau}/h$ and $\kappa = \kappa_{\text{nsf}} = \frac{1 + e^{-\tau}}{1 + e^{-(\tau + \gamma)}}$. Then

$$\log\left(\frac{1}{\varepsilon_{\text{nsf}}} - 1\right) - \log h = \log\left(\frac{1}{e^{-\tau}/h} - 1\right) - \log h = \log(he^\tau - 1) - \log h = \log(e^\tau - h^{-1}), \tag{27}$$

so the right side of the incompatibility condition equals

$$\frac{2\tau}{\log(e^\tau - h^{-1})} \xrightarrow[\tau \to \infty]{} 2. \tag{28}$$

Meanwhile,

$$\log \kappa_{\text{nsf}} = \log(1 + e^{-\tau}) - \log\left(1 + e^{-(\tau + \gamma)}\right) \tag{29}$$

$$= \log\left(1 + \frac{e^{-\tau}\left(1 - e^{-\gamma}\right)}{1 + e^{-(\tau + \gamma)}}\right) \sim e^{-\tau}\left(1 - e^{-\gamma}\right) \quad (\tau \to \infty), \tag{30}$$

hence

$$\frac{\gamma}{\log \kappa_{\text{nsf}}} \xrightarrow[\tau \to \infty]{} \infty. \tag{31}$$

Therefore, for any fixed $\gamma > 0$, the incompatibility condition holds for all sufficiently large $\tau$, so no single softmax temperature can match NSF uniformly on $\mathcal{S}(\tau, \gamma, \mathcal{H}, \mathcal{L})$. $\square$

**Remark 3** (Middle scores). *Allowing additional scores in $(-\tau, \tau)$ only strengthens the NSF suppression bound because the denominator $S$ increases, and it does not weaken the softmax lower bound equation 23 on the high to high ratio since that ratio is independent of other coordinates. The softmax low suppression supremum equation 24 is still attained by driving all non-high and non-$j$ scores to $-\infty$, so the temperature constraints remain necessary.*

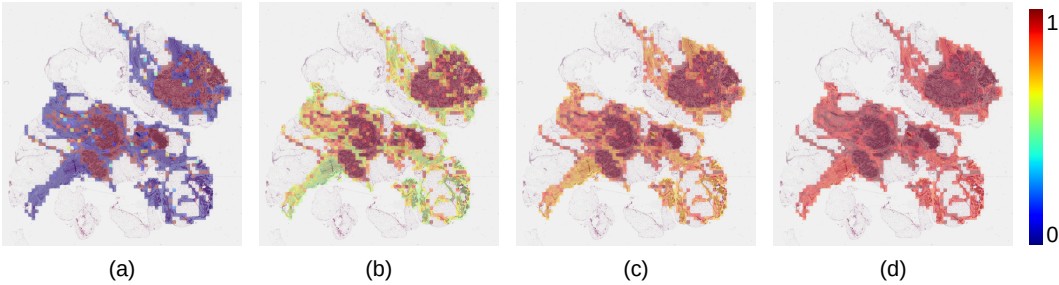

Figure 7: Ablation study of applying the softmax function with temperature scaling to the attention scores: (a) attention distribution of the proposed ASMIL, (b) softmax with $T = 2$ applied to the anchor model, (c) softmax with $T = 4$ applied to the anchor model, (d) softmax with $T = 8$ applied to the anchor model.

## F  ALTERNATIVE TO NSF IN ANCHOR MODEL

### F.1  SOFTMAX WITH TEMPERATURE SCALING

A straightforward approach to mitigating over-concentration is to apply softmax with temperature scaling (Hinton et al., 2015; Ye et al., 2024; Yang et al.; 2024; 2025). This can indeed yield less concentrated attention distribution; however, as we observe in this section, a large temperature produces an overly smooth distribution, approaching a uniform distribution. This makes all tiles nearly indistinguishable, effectively reducing the operation to mean pooling and compromising interpretability. To illustrate this, we conduct experiments on the BRACS dataset using the same training protocol as in Section 5, summarize the results in Table 4, and visualize the attention maps in Figure 7.

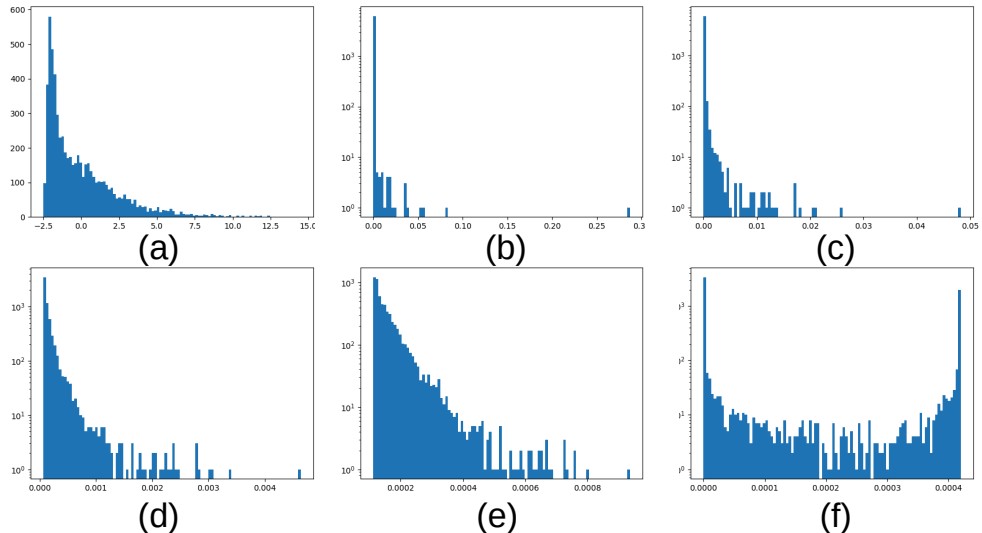

Figure 8: Histograms of (a) raw attention scores, (b) attention distribution obtained by the softmax function with temperature $T = 1$, (c) $T = 2$, (d) $T = 4$, (e) $T = 8$, and **(f) attention distribution computed using an NSF**. The Y-axis is displayed on a logarithmic scale for better visualization.

Furthermore, to clarify the differences between the NSF and softmax, we plot the histograms of the attention scores—(a) outputs from the NSF and softmax with various temperature scalings—in Figure 8. As shown, the saturation property of the NSF suppresses excessively large values.

Table 4: Subtyping performance on BRACS, when we apply softmax with temperature scaling to the anchor model.

| BRACS | | | | | | | |
|---|---|---|---|---|---|---|---|
| Normalized Sigmoid | | Softmax T=2 | | Softmax T=4 | | Softmax T=8 | |
| F1 score ↑ | AUC ↑ | F1 score ↑ | AUC ↑ | F1 score ↑ | AUC ↑ | F1 score ↑ | AUC ↑ |
| $0.781_{\pm0.042}$ | $0.914_{\pm0.014}$ | $0.667_{\pm0.049}$ | $0.860_{\pm0.027}$ | $0.712_{\pm0.029}$ | $0.876_{\pm0.012}$ | $0.688_{\pm0.037}$ | $0.858_{\pm0.031}$ |

Table 5: ASMIL performance when replacing NSF with entmax on CAMELYON-16.

| CAMELYON-16 | | | | | | |
|---|---|---|---|---|---|---|
| Metric | NSF | entmax$_{\alpha=2}$ (sparsemax) | entmax$_{\alpha=1.75}$ | entmax$_{\alpha=1.5}$ (entmax-15) | entmax$_{\alpha=1.25}$ | entmax$_{\alpha=1}$ (softmax) |
| F1 score ↑ | $0.965_{\pm0.020}$ | $0.938_{\pm0.031}$ | $0.927_{\pm0.034}$ | $0.937_{\pm0.014}$ | $0.910_{\pm0.026}$ | $0.942_{\pm0.0147}$ |
| AUC ↑ | $0.985_{\pm0.017}$ | $0.964_{\pm0.012}$ | $0.959_{\pm0.017}$ | $0.960_{\pm0.017}$ | $0.925_{\pm0.031}$ | $0.963_{\pm0.020}$ |
| Time per epoch ↓ | 6.340s | 8.451s | 8.452s | 8.451s | 8.450s | 6.336s |

## F.2 ENTMAX

Entmax is a family of mappings that convert a score vector $z \in \mathbb{R}^d$ into a probability vector $p \in \Delta^d$ by maximizing a linear score plus Tsallis-$\alpha$ entropy Tsallis (1988) $H_\alpha^T$:

$$\text{entmax}_\alpha(z) = argmax_{p \in \Delta^d} z^T p + H_\alpha^T(p), \tag{32}$$

The solution admits a closed form

$$\boldsymbol{\alpha}_i = \left[\frac{\alpha-1}{\alpha}(z_i - \tau)\right]_+^{\frac{1}{\alpha-1}}, \text{with} \sum_i \boldsymbol{\alpha}_i = 1, \tag{33}$$

where $\tau$ is a threshold chosen so that the probabilities sum to one. As limiting cases, $\alpha \to 1$, yields softmax, and while $\alpha = 2$ yields sparsemax (Martins & Astudillo, 2016).

While entmax offers controllable sparsity, two drawbacks are pertinent to MIL-based WSI analysis: (*i*) **Lack of selective flattening**, entmax is monotone in $z$ on its active support and does not explicitly equalize top-probability entries. (*ii*) **Higher computational cost**. Computing $\tau$ in Equation (33) requires the bisection method, which adds non-trivial overhead vs. NSF's fully closed-form normalization. These differences matter for MIL on WSIs, where multiple correlated tumor foci can be present: we prefer mechanisms that both discourage over-peaky attention and keep computation predictable. We replaced NSF with $entmax_\alpha$ inside ASMIL and swept $\alpha \in \{2, 1.75, 1.5, 1.25, 1\}$. For $\alpha = 1$, we used PyTorch softmax; for $\alpha > 1$, we solved for $\tau$ via bisection. The implementation follows the reference code from DeepSPIN[5]. All other hyperparameters, model, and data pipeline were kept fixed. We report results on CAMELYON-16 in Table 5. As seen, across $\alpha$, entmax underperforms NSF on both F1 and AUC and incurs a $33.5\%$ increase in epoch time vs. NSF.

## G  APPLYING NORMALIZED SIGMOID TO THE ONLINE MODEL

One might question the rationale behind applying the NSF to the anchor model while using the softmax function for the online model during training. To investigate this design choice, we experiment with applying the NSF to both the online and anchor models and evaluate the model's subtyping performance on the CAMELYON-16 and BRACS datasets. The results, presented in Table 6, reveal a F1 score drop of over 6% on the BRACS dataset. We attribute this degradation to the inherent characteristics of the sigmoid function: when it saturates, its gradients diminish, leading to vanishing gradients in the attention mechanism and thereby impairing the learning process.

To further investigate the potential of applying NSF in the online model, we consider the following mixed attention variant:

$$\alpha_i'(z) = \zeta \alpha_i^{\text{SMX}}(z) + (1 - \zeta) \alpha_i^{\text{NSF}}(z), \tag{34}$$

where $\zeta = \sigma(\xi)$ and $\xi$ is a trainable scalar that balances the contributions of the softmax and NSF mappings, initialized with $\xi = 0$. We evaluate this variant on CAMELYON-16, CAMELYON-17,

---

[5]https://github.com/deep-spin/entmax

Table 6: Ablation study on the impact of applying the normalized sigmoid (NS) function to both the online and anchor models. ✓ indicates that NSF is applied to both models, while ✗ denotes the default setting where NSF is applied only to the anchor model. Subtyping performance is evaluated on the CAMELYON-16 and BRACS datasets using F1 score and AUC. A significant performance drop is observed on CAMELYON-16 when NSF is applied to both models.

| Dataset | CAMELYON-16 | |
|---|---|---|
| Online NSF | F1 score ↑ | AUC ↑ |
| ✓ | $0.920_{\pm 0.020}$ | $0.936_{\pm 0.021}$ |
| ✗ | $0.965_{\pm 0.020}$ | $0.985_{\pm 0.017}$ |
| Dataset | BRACS | |
| Online NSF | F1 score ↑ | AUC ↑ |
| ✓ | $0.726_{\pm 0.014}$ | $0.865_{\pm 0.017}$ |
| ✗ | $0.781_{\pm 0.042}$ | $0.914_{\pm 0.014}$ |

Table 7: Comparison between ASMIL trained with the standard softmax function in the online model (ASMIL w. Softmax) and with the mixed attention function defined in Equation (34) (ASMIL w. Mixture). The more flexible trainable mapping does not yield improvements over the simpler softmax baseline.

| Dataset | CAMELYON-16 | | CAMELYON-17 | | BRACS | |
|---|---|---|---|---|---|---|
| Mertic | F1 score | AUC | F1 score | AUC | F1 score | AUC |
| ASMIL W. SoftMax | $0.965_{\pm 0.020}$ | $0.985_{\pm 0.017}$ | $0.689_{\pm 0.005}$ | $0.898_{\pm 0.010}$ | $0.781_{\pm 0.042}$ | $0.914_{\pm 0.014}$ |
| ASMIL W. Mixture | $0.953_{\pm 0.023}$ | $0.972_{\pm 0.030}$ | $0.686_{\pm 0.012}$ | $0.889_{\pm 0.009}$ | $0.774_{\pm 0.054}$ | $0.910_{\pm 0.067}$ |
| $\zeta$ in Equation (34) | 0.9952 | | 0.9894 | | 0.9963 | |

and BRACS, and report the results in Table 7. The mixed mapping does not outperform the default softmax, and the learned $\zeta$ consistently converges to values close to one, indicating that the online model prefers softmax, which does not suffer from gradient-vanishing issues.

## H    ALTERNATIVE STABILIZATION METHODS AND WHY THE ANCHOR IS PREFERABLE

Let $\boldsymbol{\alpha}_t(x) \in \Delta^N$ denote the attention distribution for slide $x$ at epoch $t$, obtained from scores $\boldsymbol{z}_t(x) \in \mathbb{R}^N$. We diagnose instability by the Jensen-Shannon divergence

$$\mathrm{JSD}_t(x) = \mathrm{JSD}\big(\boldsymbol{\alpha}_t(x) \| \boldsymbol{\alpha}_{t-1}(x)\big), \tag{35}$$

which we empirically find remains high when training attention-based MIL with only bag-level labels. We present a natural alternative that targets this instability and explain why the anchor model is preferred.

### H.1    ALTERNATIVE: PER-SLIDE TEMPORAL ENSEMBLING OF ATTENTION

Maintain a per slide exponential moving average (EMA) of past attentions and penalize deviation from it:

$$\tilde{\boldsymbol{\alpha}}_t(x) = \rho\,\tilde{\boldsymbol{\alpha}}_{t-1}(x) + (1-\rho)\,\boldsymbol{\alpha}_t(x),\ \rho \in (0,1); \qquad \mathcal{L}_{\mathrm{AS}}(x) = \mathsf{KL}\Big(\boldsymbol{\alpha}_t(x) \| \mathsf{sg}\big(\tilde{\boldsymbol{\alpha}}_t(x)\big)\Big). \tag{36}$$

The EMA target changes slowly when $\rho$ is close to one, which directly shrinks epoch-to-epoch drift of $\boldsymbol{\alpha}_t$ and reduces $\mathsf{JSD}(\boldsymbol{\alpha}_t \| \boldsymbol{\alpha}_{t-1})$. *However,*

($i$) It has to maintain a length-$N$ vector per slide. For $S$ slides and average $\bar{N}$ tiles, memory is $O(S\bar{N})$ floats, which can be substantial for gigapixel WSIs and prevent scaling to larger datasets. ($ii$) The EMA target still uses softmax normalization, which cannot achieve selective flattening across informative tokens; see Theorem 1.

## H.2 WHY ASMIL'S ANCHOR IS PREFERABLE

We highlight two main reasons for using an anchor model to stabilize the attention distribution rather than relying on temporal ensembling.

**NSF provides selective flattening that softmax cannot match.**

Replacing softmax with the normalized sigmoid function (NSF) in the anchor yields $\boldsymbol{\alpha}^{\text{nsf}}(x)$, which equalizes probabilities among truly high-score tiles while suppressing low-score ones. By Theorem 1, no single softmax temperature can realize both behaviors across a broad class of score vectors. Consequently, methods that retain softmax-based targets inherit these limitations.

**Memory and implementation simplicity.**

The anchor-based approach adds only one extra forward pass and maintains an exponential moving average (EMA) of the anchor parameters during training. It does not require storing per-slide attention distributions, making the approach scalable to large WSI datasets.

Thus, an anchor model is preferable for scalable training on large MIL datasets and for preventing attention over-concentration.

# I WHY MATCHING THE TEACHER (ANCHOR) MODEL'S SOFTMAX FEATURE VECTOR CANNOT STABILIZE THE ATTENTION DISTRIBUTION

Table 8: Ratio of affinely dependent feature bags in the CAMELYON-16, CAMELYON-17, and BRACS datasets; most bags are affinely dependent.

| Dataset | CAMELYON-16 | CAMELYON-17 | BRACS |
|---|---|---|---|
| The ratio of affine dependent feature bags | 99.24% | 99.80% | 96.08% |

In this section, we show why matching the softmax of the bag-level feature is a suboptimal strategy for stabilizing attention distributions. To this end, we prove that recovering the attention vector $\boldsymbol{\alpha}$ by matching $\text{softmax}(\alpha^T X)$ is, in general, ill-posed: the map $f : \Delta^K \to \Delta^d$, defined by $f(\alpha) = \text{softmax}(\alpha^T X)$ with $X \in \mathbb{R}^{K \times d}$, fails to be injective when the feature matrix $X$ is affinely dependent.

*Proof.* Assume the rows $x_1, \ldots, x_K \in \mathbb{R}^d$ of $X$ are affinely dependent. By definition there exists a nonzero vector $\psi \in \mathbb{R}^K$ such that

$$\sum_{i=1}^{K} \psi_i = 0 \quad \text{and} \quad \sum_{i=1}^{K} \psi_i x_i = 0.$$

Let $\alpha \in \Delta^K$ be any probability vector and choose $\epsilon > 0$ small enough that $\alpha' = \alpha + \epsilon\psi$ satisfies $\alpha'_i \geq 0$ for every $i$. Note $\sum_i \alpha'_i = \sum_i \alpha_i + \epsilon \sum_i \psi_i = 1$, so $\alpha' \in \Delta^K$. Since $\sum_{i=1}^{K} \psi_i x_i = 0$ we have

$$(\alpha')^T X = \alpha^T X + \epsilon\psi^T X = \alpha^T X.$$

Therefore

$$f(\alpha') = \text{softmax}((\alpha')^T X) = \text{softmax}(\alpha^T X) = f(\alpha).$$

Because $\psi \neq 0$ and $\epsilon \neq 0$ we have $\alpha' \neq \alpha$, hence $f$ is not injective. $\square$

Thus, matching the softmax of the bag feature cannot reliably recover or stabilize the attention distributions when the feature bag is affinely dependent. Table 8 confirms that most feature bags extracted by VIT-S (Kang et al., 2023) from WSI datasets are indeed affinely dependent.

Table 9: The F1 score and AUC of different MIL approaches on two WSI subtyping datasets.

| | PathGen-Clip-VIT-L | | | |
|---|---|---|---|---|
| Dataset | CAMELYON-16 | | CAMELYON-17 | |
| Method | F1 score ↑ | AUC ↑ | F1 score ↑ | AUC ↑ |
| Clam-SB | $0.941_{\pm0.014}$ | $0.960_{\pm0.015}$ | $0.622_{\pm0.031}$ | $0.899_{\pm0.012}$ |
| TransMIL | $0.951_{\pm0.024}$ | $0.968_{\pm0.028}$ | $0.656_{\pm0.021}$ | $0.892_{\pm0.014}$ |
| DSMIL | $0.895_{\pm0.038}$ | $0.949_{\pm0.017}$ | $0.582_{\pm0.062}$ | $0.887_{\pm0.013}$ |
| IBMIL | $0.935_{\pm0.014}$ | $0.953_{\pm0.009}$ | $0.629_{\pm0.027}$ | $0.884_{\pm0.016}$ |
| MHIM-MIL | $0.946_{\pm0.33}$ | $0.984_{\pm0.016}$ | $0.594_{\pm0.090}$ | $0.912_{\pm0.009}$ |
| ABMIL | $0.953_{\pm0.018}$ | $0.972_{\pm0.010}$ | $0.610_{\pm0.025}$ | $0.864_{\pm0.017}$ |
| AEM | $0.967_{\pm0.025}$ | $0.988_{\pm0.013}$ | $0.688_{\pm0.016}$ | $0.905_{\pm0.005}$ |
| ASMIL | $0.974_{\pm0.021}$ | $0.990_{\pm0.014}$ | $0.699_{\pm0.020}$ | $0.929_{\pm0.016}$ |
| | UNI-VIT-L | | | |
| Method | F1 score ↑ | AUC ↑ | F1 score ↑ | AUC ↑ |
| ABMIL | $0.968_{\pm0.011}$ | $0.996_{\pm0.003}$ | $0.605_{\pm0.047}$ | $0.885_{\pm0.015}$ |
| AEM | $0.975_{\pm0.003}$ | $0.998_{\pm0.003}$ | $0.633_{\pm0.024}$ | $0.863_{\pm0.017}$ |
| ASMIL | $0.980_{\pm0.004}$ | $0.998_{\pm0.002}$ | $0.672_{\pm0.035}$ | $0.866_{\pm0.014}$ |

## J  APPLYING ASMIL TO FEATURES EXTRACTED BY A WSI FOUNDATION MODEL

In recent years, foundation models have enabled strong open-source feature extractors that markedly improve the performance of computational-pathology systems. To assess the generalizability of our approach, we apply ASMIL to features produced by two such extractors, UNI Chen et al. (2024) and PATHGEN-clip Sun et al. (2025), for the subtyping task on the CAMELYON-16 and CAMELYON-17 datasets. As reported in Table 9, ASMIL consistently outperforms all baseline methods when used with features extracted by foundation models, yielding the best F1 and AUC.

## K  ABLATION STUDY

### K.1  ABLATION OF THE COEFFICIENT $\beta$

The coefficient $\beta > 0$ in Equation (9) balances the stabilization and classification objectives. To assess its impact on final performance, we sweep $\beta \in \{0, 0.1, 0.25, 0.5, 0.75, 1.0, 1.5, 2.0, 2.5, 4, 5\}$ on the CAMELYON-16 and BRACS datasets. Except for $\beta$, all experimental settings are identical to those in Section 5.1. We report F1 score and AUC in Figures 9 and 10; results are averaged over five random seeds. Overall, model performance is relatively insensitive to the choice of $\beta$: both

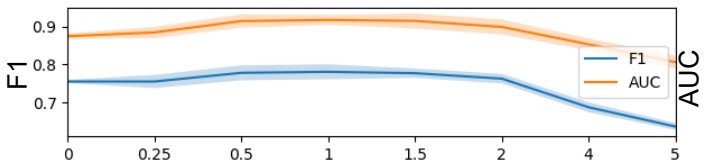

Figure 9: Ablation study on the coefficient $\beta$, on CAMELYON-16 dataset.

F1 score and AUC plateau for $\beta \in [0.5, 1.5]$. Accordingly, we set $\beta = 1$ as the default for all experiments.

### K.2  ABLATION STUDY ON NUMBER OF TRAINABLE FEAT TOKENS

In this section, we investigate how varying the number of trainable tokens influences model performance. To this end, we sweep a number of trainable tokens in the range of $[2, 4, 8, 16]$, and report the corresponding accuracy on CAMELYON-16, CAMELYON-17, and BRACS in Table 10 In the

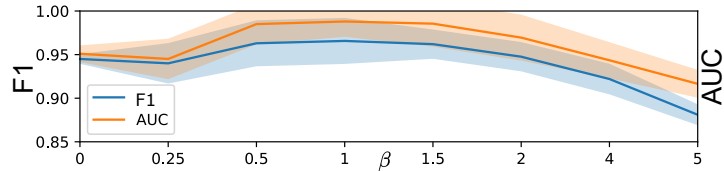

Figure 10: Ablation study on the coefficient $\beta$, on BRACS dataset.

Table 10: Ablation results on the number of tokens on different WSI datasets.

| # FEAT tokens | 2 | 4 | 8 | 16 |
|---|---|---|---|---|
| **CAMELYON-16** | | | | |
| F1 score $\uparrow$ | $0.930_{\pm 0.012}$ | $0.946_{\pm 0.009}$ | $0.965_{\pm 0.012}$ | $0.960_{\pm 0.006}$ |
| AUC $\uparrow$ | $0.932_{\pm 0.017}$ | $0.973_{\pm 0.011}$ | $0.985_{\pm 0.013}$ | $0.981_{\pm 0.009}$ |
| **CAMELYON-17** | | | | |
| F1 score $\uparrow$ | $0.556_{\pm 0.012}$ | $0.610_{\pm 0.009}$ | $0.674_{\pm 0.016}$ | $0.689_{\pm 0.005}$ |
| AUC $\uparrow$ | $0.784_{\pm 0.019}$ | $0.833_{\pm 0.011}$ | $0.879_{\pm 0.024}$ | $0.898_{\pm 0.010}$ |
| **BRACS** | | | | |
| F1 score $\uparrow$ | $0.721_{\pm 0.009}$ | $0.766_{\pm 0.012}$ | $0.781_{\pm 0.004}$ | $0.782_{\pm 0.004}$ |
| AUC $\uparrow$ | $0.871_{\pm 0.004}$ | $0.903_{\pm 0.014}$ | $0.914_{\pm 0.004}$ | $0.912_{\pm 0.026}$ |

experiment, we apply 8 trainable tokens for CAMELYON-16 and BRACS, and 16 trainable tokens on the CAMELYON-17 dataset.

### K.3 ABLATION ON ANCHOR MODEL UPDATE

### K.3.1 EFFECT OF ANCHOR MODEL UPDATE FREQUENCY

Table 11: Ablation study on anchor model update frequency, where batch-wise updates consistently outperform epoch-wise updates in both F1 score and AUC on BRACS and CAMELYON-16.

| Dataset | BRACS | |
|---|---|---|
| Update | F1 score $\uparrow$ | AUC $\uparrow$ |
| Epoch | $0.742_{\pm 0.015}$ | $0.871_{\pm 0.003}$ |
| Batch | $0.781_{\pm 0.042}$ | $0.914_{\pm 0.014}$ |
| Dataset | CAMELYON-16 | |
| Update | F1 score $\uparrow$ | AUC $\uparrow$ |
| Epoch | $0.920_{\pm 0.020}$ | $0.936_{\pm 0.021}$ |
| Batch | $0.965_{\pm 0.020}$ | $0.984_{\pm 0.017}$ |

To assess the impact of anchor update frequency, we compare epoch-wise and batch-wise update strategies on BRACS and CAMELYON-16 (Table 11). The results show that batch-wise updates consistently deliver superior performance. On BRACS, batch-wise updates improve the F1 score by 3.9% and the AUC by 4.9%. On CAMELYON-16, the improvement is even more substantial, with the F1 score increasing by 4.9% and the AUC by 5.1%. These gains confirm that frequent updates enable the anchor model to provide a stable and closely aligned attention reference for the online model, leading to better performance.

### K.4 IMPACT OF THE RANDOM DROP RATE

We evaluated the effect of random token dropping on model performance using CAMELYON-16 and BRACS, measuring both F1 and AUC across several trainable-token budgets. Results in Figure 11 show a consistent trend: performance rises from low $B$, peaks around $B = 0.5$, then degrades

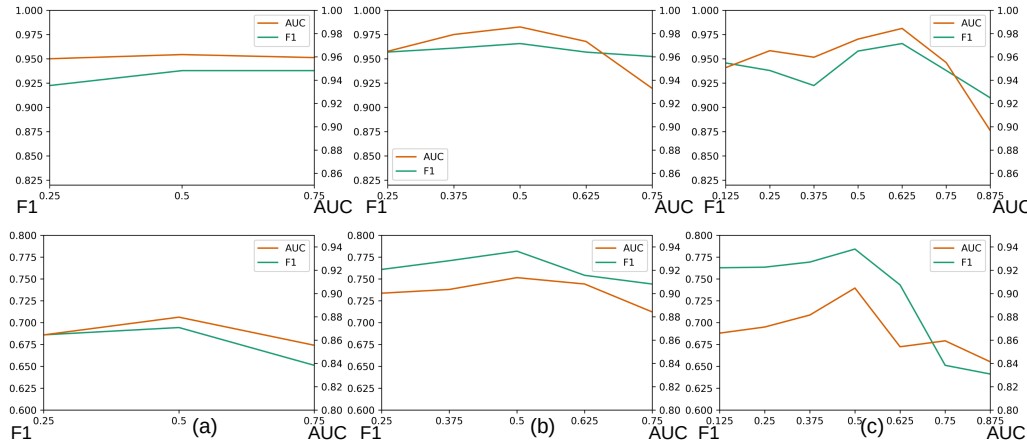

Figure 11: Ablation study of random drop probability ($B$) vs. model F1 score and AUC on CAMELYON-16 (top row) and BRACS (bottom row). Across both datasets and trainable-token settings (a) 4 tokens, (b) 8 tokens, and (c) 16 tokens, the test F1 score and AUC consistently peak around $B = 0.5$.

for larger values. This pattern holds across datasets and capacities, indicating a stable trade-off between regularization and information loss.

Mechanistically, moderate token dropping (0.4–0.7) provides useful regularization, encouraging robustness to missing context and reducing overfitting to redundant or spurious tiles, while excessive dropping increases the chance of discarding diagnostically critical patches and thus harms recall and ranking. We therefore recommend tuning $B$ in the range of $0.4 - 0.6$. In Appendix K.5 we plot test F1 score and AUC across training epochs to demonstrate that random token dropping mitigates overfitting.

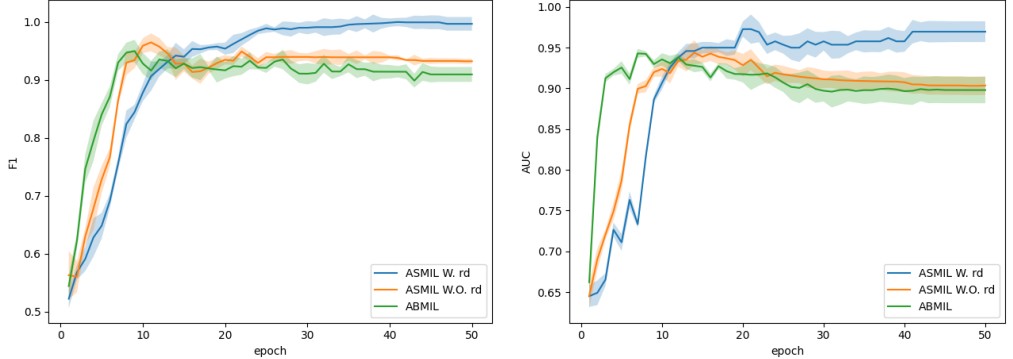

Figure 12: Performance comparison between ABMIL (Ilse et al., 2018), ASMIL with random drop (ASMIL W. rd), and ASMIL without random drop (ASMIL w/o rd). Both ABMIL and ASMIL w/o rd show signs of overfitting, as their F1 score and AUC peak and then decline. In contrast, ASMIL with random drop maintains stable performance across training, demonstrating that random drop effectively mitigates overfitting.

## K.5    RANDOM DROP MITIGATES OVERFITTING

To verify that random drop is an efficient regularizer for attention-based MIL on WSIs, we trained three variants on CAMELYON-16: ($i$) ABMIL, ($ii$) ASMIL without random drop, and ($iii$) ASMIL with random drop (ours) with $B = 0.5$. The figure reports validation F1 and AUC over training epochs.

As shown in the Figure 12, both ABMIL and ASMIL without random drop exhibit overfitting: F1 score and AUC rise early, peak, and then decline with continued training. In contrast, ASMIL with random drop maintains high and stable F1/AUC throughout later epochs, with noticeably reduced

Table 12: Statistical comparison of ASMIL with and without the anchor model. We report the mean performance over 10 random seeds along with p-values from DeLong tests for AUC and permutation tests for F1.

| Dataset | Model | AUC | F1 | $p_{\text{AUC}}$ | $p_{\text{F1}}$ |
|---|---|---|---|---|---|
| CAMELYON-16 | w/o anchor | 0.942 | 0.979 | 0.013 | 0.024 |
| | w/ anchor | 0.967 | 0.983 | | |
| CAMELYON-17 | w/o anchor | 0.642 | 0.879 | 0.024 | 0.035 |
| | w/ anchor | 0.693 | 0.899 | | |
| BRACS | w/o anchor | 0.729 | 0.866 | 0.012 | 0.009 |
| | w/ anchor | 0.784 | 0.916 | | |

run-to-run variability (shaded regions). These trajectories empirically validate that random drop curbs the late-epoch degradation that accompanies weak supervision on CAMELYON-16. This observation aligns with our analysis that overfitting is a recurring failure mode for attention-based MIL on WSI datasets.

### K.6    SIGNIFICANCE TEST ON THE EFFECT OF ONLINE MODEL

To assess whether the performance gains from the anchor model are statistically meaningful, we perform paired significance tests between ASMIL with and without the anchor over multiple 10 seeds. For AUC, we apply DeLong's test, and for F1, we use a non-parametric permutation test. Across CAMELYON-16, CAMELYON-17, and BRACS, the anchor-augmented ASMIL consistently achieves higher AUC and F1 than its non-anchor counterpart, and these improvements are statistically significant ($p < 0.05$) for both metrics on each dataset (see Table 12).

## L    QUANTITATIVE LOCALIZATION RESULTS AND ADDITIONAL VISUALIZATION

Predicted masks are generated as follows. For attention-based methods (CLAM (Lu et al., 2021), TransMIL (Shao et al., 2021), DTFD-MIL (Zhang et al., 2022), DSMIL (Li et al., 2021b) and CAMIL Fourkioti et al. (2024)), we use the tile-level attention distribution. For ASMIL, the per-tile attention distribution is computed by averaging the attention distributions from all FEAT tokens to that tile. Unless otherwise noted, we rescale all per-tile scores to $[0, 1]$ and threshold at $0.5$ to produce binary masks across all methods.

For tumor localization on CAMELYON-16, we follow the official challenge protocol and report the lesion-level Free-Response ROC (FROC) (Miller, 1969; Bunch, 1978; Zhang et al., 2025a). Concretely, model outputs are converted to point detections; a detection is counted as a true positive if it lies within 75 μm of any ground-truth tumor region (implemented in the official script via a distance-transform "evaluation mask"), otherwise it is a false positive. We then sweep the detection score threshold to trace sensitivity versus the average number of false positives per normal WSI, and compute the standard CAMELYON-16 FROC score as the mean sensitivity at 0.25, 0.5, 1, 2, 4, 8 FP/WSI.

Quantitative results for FROC, Dice, and specificity are reported in Table 13, ASMIL achieves the best FROC and Dice on cancerous slides and higher specificity on normal slides, yielding fewer false positives and more contiguous lesion maps compared to baselines.

Figure 13 presents additional visualizations on the CAMELYON-16 dataset. It shows ASMIL attention maps for tumor slides containing both small and large cancerous regions; rows 1 and 3 provide the ground-truth annotations, and rows 2 and 4 show the corresponding attention maps.

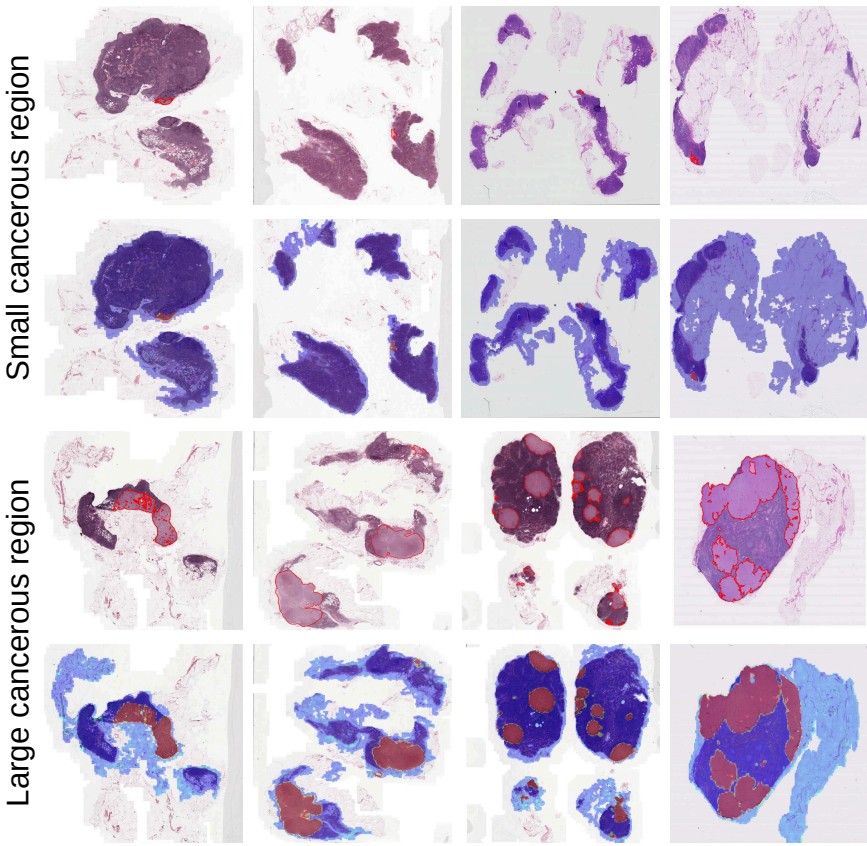

Figure 13: Additional qualitative examples of tumor regions and ASMIL attention maps. Rows 1 and 3 show the ground-truth tumor annotations (cancerous regions outlined in red), and rows 2 and 4 show the corresponding ASMIL attention maps.

Table 13: Localization results on CAMELYON-16.

| Method | Dice ↑ | Specificity ↑ | FROC ↑ |
|---|---|---|---|
| CLAM-SB | 0.459 | 0.987 | 0.4257 |
| TransMIL | 0.103 | 0.999 | 0.0866 |
| DTFD-MIL | 0.525 | 0.999 | 0.4712 |
| DSMIL | 0.259 | 0.863 | 0.4506 |
| CAMIL | 0.515 | 0.980 | 0.4612 |
| ASMIL | 0.586 | 0.999 | 0.4941 |

## M  COMPUTATIONAL COST

This section reports the computational cost of ASMIL, as well as the additional cost incurred when integrating the anchor model into the baseline methods.

### M.1  COMPARISON OF THE COMPUTATIONAL COST BETWEEN ASMIL AND BASELINE METHODS

We conducted a detailed evaluation of the computational overhead introduced by our proposed AS-MIL framework, focusing on three primary metrics: floating-point operations (FLOPs), training time per epoch, and peak memory consumption. All experiments were executed under uniform hardware conditions, specifically a single NVIDIA RTX 5000 GPU coupled with an Intel Xeon W-2265 CPU and 64 GB of RAM, ensuring a fair comparison across methods.

Table 14: Computational cost on BRACS (lower is better). We report training time and peak memory per epoch, and inference FLOPs, latency, and memory. ASMIL (ours) delivers efficient inference, cutting compute by 30.6%, latency by 29.2%, and memory by 20.3% compared with TransMIL, while requiring $4\times$ less training memory than MHIM-MIL.

| BRACS | Training | | | | |
|---|---|---|---|---|---|
| Method | CLAM-SB | ABMIL | TransMIL | MHIM-MIL | ASMIL |
| Time | 2.26s | 0.95s | 5.99s | 19.4s | 7.49s |
| Memory | 94MB | 90MB | 340MB | 2178MB | 570MB |
| BRACS | Inference | | | | |
| FLOPs | 162M | 164M | 781M | 345M | 542M |
| Time | 0.45s | 0.37s | 0.74s | 0.40s | 0.52s |
| Memory | 69MB | 39MB | 246MB | 61MB | 196MB |

Table 15: Inference FLOPs, training time per epoch (Time), and memory usage (Memory) for four well-known methods, CLAM-SB, TransMIL, DSMIL, and ABMIL, with and without the anchor model. The anchor model incurs only minor computational overhead. FLOPs are measured using a fixed bag size of 2000 instances.

| BRACS | Training | | Inference | | |
|---|---|---|---|---|---|
| Method | Time | Memory | FLOPs | Time | Memory |
| CLAM-SB    w/o anchor | 2.26s | 94MB | 162M | 0.45s | 69MB |
| CLAM-SB W. anchor | 2.69s | 120MB | 162M | 0.45s | 69MB |
| TransMIL    w/o anchor | 5.99s | 340MB | 781M | 0.74s | 246MB |
| TransMIL W. anchor | 7.27s | 443MB | 781M | 0.74s | 246MB |
| DSMIL    w/o anchor | 0.57s | 60 MB | 103M | 1.09s | 113 MB |
| DSMIL W. anchor | 0.58s | 145MB | 103M | 1.09s | 113 MB |
| ABMIL    w/o anchor | 0.95s | 90MB | 164M | 0.37s | 39MB |
| ABMIL W. anchor | 1.17s | 162MB | 164M | 0.37s | 39MB |

During training, ASMIL demonstrates a competitive balance between efficiency and computational demand. On average, ASMIL requires 542M FLOPs per batch, which is lower than MHIM-MIL. The training time per epoch for ASMIL is 7.49s, substantially faster than MHIM-MIL (19.4s) and comparable to TransMIL (5.99s), while remaining higher than ABMIL and CLAM-SB. In terms of peak memory usage, ASMIL consumes 570 MB, markedly lower than MHIM-MIL (2178 MB). These results indicate that ASMIL maintains a favorable computational profile, offering a scalable alternative to more resource-intensive methods.

In inference, ASMIL continues to show strong efficiency. It requires 542M FLOPs, substantially fewer than TransMIL and comparable to MHIM-MIL. Inference time for ASMIL is 0.52s per epoch, slightly slower than CLAM-SB (0.45s) but faster than TransMIL. Peak memory usage during inference is 196 MB, markedly lower than TransMIL, highlighting ASMIL's efficient memory footprint relative to its computational performance. Overall, ASMIL delivers high-performance multiple-instance learning while keeping computational cost affordable.

## M.2    ADDITIONAL COMPUTATIONAL COST INTRODUCED BY ANCHOR MODEL

We conducted a detailed evaluation of the computational overhead introduced by integrating the anchor model into four widely used MIL methods, namely CLAM-SB, TransMIL, DSMIL, and ABMIL, all measured on the BRACS dataset. The results are summarized in Table 15.

Because no gradients are computed through the anchor model, and only the attention layer is updated during training, the computational overhead is small. As shown in Table 15, integrating the anchor model into CLAM-SB, TransMIL, DSMIL, and ABMIL introduces only a modest increase in training time and memory usage, while the FLOPs remain unchanged. For example, training time for CLAM-SB increases from 2.26s to 2.69s and memory usage from 94 MB to 120 MB, with larger models like TransMIL showing slightly higher overhead. Importantly, during inference, the

Table 16: C-index for WSI-based survival prediction using vision-only MIL models.

| Method | BLCA | BRCA | GBMLGG | LUAD | UCEC |
|--------|------|------|--------|------|------|
| ABMIL ICML 2018 | $0.5581_{\pm 0.031}$ | $0.5825_{\pm 0.035}$ | $0.7935_{\pm 0.032}$ | $0.6121_{\pm 0.050}$ | $0.6667_{\pm 0.033}$ |
| TransMIL NeurIPS 2021 | $0.5885_{\pm 0.055}$ | $0.6140_{\pm 0.060}$ | $0.7956_{\pm 0.015}$ | $0.5708_{\pm 0.050}$ | $0.6380_{\pm 0.067}$ |
| ILRA ICLR 2023 | $0.5549_{\pm 0.053}$ | $0.5705_{\pm 0.067}$ | $0.7742_{\pm 0.014}$ | $0.5179_{\pm 0.081}$ | $0.6503_{\pm 0.064}$ |
| $R^2$T-MIL CVPR 2024 | $0.5775_{\pm 0.024}$ | $0.5476_{\pm 0.095}$ | $0.7757_{\pm 0.024}$ | $0.5711_{\pm 0.076}$ | $0.6510_{\pm 0.087}$ |
| DeepAttnMISL MIA 2020 | $0.5646_{\pm 0.035}$ | $0.5346_{\pm 0.036}$ | $0.6750_{\pm 0.048}$ | $0.4678_{\pm 0.039}$ | $0.6259_{\pm 0.086}$ |
| Patch-GCN MICCAI 2021 | $0.6124_{\pm 0.031}$ | $0.6375_{\pm 0.033}$ | $0.7999_{\pm 0.021}$ | $0.5922_{\pm 0.053}$ | $0.7212_{\pm 0.025}$ |
| ASMIL (Ours) | $0.6133_{\pm 0.047}$ | $0.6396_{\pm 0.044}$ | $0.8036_{\pm 0.018}$ | $0.6001_{\pm 0.093}$ | $0.7243_{\pm 0.0488}$ |

anchor model is discarded, resulting in identical FLOPs, execution time, and memory consumption compared to the baseline methods. These results demonstrate that the anchor model provides performance benefits during training with minimal computational cost and does not affect deployment efficiency, making it an effective and practical addition to existing MIL frameworks.

## N  SURVIVAL PREDICTION

To assess whether ASMIL is also beneficial for prognosis, we extend ASMIL from slide-level classification to discrete-time overall survival prediction on histopathology WSIs. Following (Liu et al., 2025), we apply an incidence-based discrete survival formulation, *i.e.*, the survival times are mapped to $C$ non-overlapping time intervals, and the model outputs a discrete distribution over first-event times.

We follow the experimental setup of (Liu et al., 2025), and evaluate on five TCGA datasets, namely BLCA, BRCA, LUAD, and UCEC. We use the concordance index (C-index) to evaluate the model's performance; specifically, it measures how often the model assigns a higher risk score to a patient who experiences the event earlier. Formally, with a little abuse of notations, let $t_i, \delta_i, \hat{R}_i$ denote the observed time, event indicator, and predicted risk for patient $i$, the C-index is defined as

$$\text{CI} = \frac{\sum_{i,j} \mathbf{1}[t_i < t_j] \mathbf{1}[\hat{R}_i > \hat{R}_j] \delta_i}{\sum_{i,j} \mathbf{1}[t_i < t_j] \delta_i}, \tag{37}$$

where $\mathbf{1}[\cdot]$ is the indicator function. A value of CI $= 0.5$ corresponds to a random ranking, and larger values indicate better risk discrimination Yang & Ye (2024); Hamidi & Ye (2024; 2025). Following Liu et al. (2025), we compare ASMIL against six vision-only WSI survival prediction methods, namely ABMIL (Ilse et al., 2018), TransMIL (Shao et al., 2021), ILRA Xiang & Zhang (2023), $R^2$T-MIL (Tang et al., 2024), DeepAttnMISL (Yao et al., 2020), and Patch-GCN (Chen et al., 2021), all implemented on top of the same CONCH-derived patch features (Lu et al., 2023).

Table 16 reports the C-index on each TCGA dataset. ASMIL achieves the highest mean C-index among all vision-only baselines. These results indicate that stabilizing slide-level attention not only improves weakly supervised classification but also yields stronger prognostic discrimination in survival analysis.

## O  EVALUATE ASMIL OVER NON-WSI DATASET

Table 17: MIL dataset statistics.

| Dataset | Domain | Bags (pos/neg) | Total instances | Dim./inst. |
|---------|--------|----------------|-----------------|------------|
| MUSK1 | Drug activity | 92 (47/45) | 476 | 166 |
| MUSK2 | Drug activity | 102 (39/63) | 6598 | 166 |
| TIGER | Images (Blobworld segments) | 200 (100/100) | 1220 | 230 |
| FOX | Images (Blobworld segments) | 200 (100/100) | 1320 | 230 |
| ELEPHANT | Images (Blobworld segments) | 200 (100/100) | 1391 | 230 |

To demonstrate ASMIL's applicability beyond WSI, we evaluate it on five classic multiple-instance learning (MIL) benchmarks: *MUSK1* Chapman & Jain (1994a) and *MUSK2* Chapman & Jain (1994b), where each bag is a molecule and instances are its low-energy 3D conformations described by 166 attributes (a bag is positive if at least one conformation is active); and the image MIL datasets *TIGER*, *FOX*, and *ELEPHANT* Andrews et al. (2002), where each bag is a Corel image segmented into "Blobworld" regions (instances) with 230-D color/texture/shape features (a bag is positive if at least one segment contains the named animal). Standard size statistics are reported in Table 17.

Table 18: Results on the small MIL benchmark datasets.

| Methods | MUSK1 | MUSK2 | FOX | TIGER | ELEPHANT |
|---|---|---|---|---|---|
| ABMIL ICML 2018 | $0.916_{\pm 0.118}$ | $0.928_{\pm 0.109}$ | $0.952_{\pm 0.051}$ | $0.953_{\pm 0.042}$ | $0.969_{\pm 0.036}$ |
| DSMIL CVPR 2021b | $0.959_{\pm 0.053}$ | $0.952_{\pm 0.066}$ | $0.939_{\pm 0.060}$ | $0.951_{\pm 0.053}$ | $\mathbf{0.989_{\pm 0.023}}$ |
| TransMIL NeurIPS 2021 | $0.927_{\pm 0.093}$ | $0.877_{\pm 0.127}$ | $0.944_{\pm 0.050}$ | $0.963_{\pm 0.042}$ | $0.979_{\pm 0.030}$ |
| DEMIL NeurIPS 2023a | $0.963_{\pm 0.073}$ | $0.961_{\pm 0.057}$ | $0.941_{\pm 0.047}$ | $0.965_{\pm 0.035}$ | $0.969_{\pm 0.034}$ |
| RGMIL Neurips2023 | $0.968_{\pm 0.060}$ | $0.963_{\pm 0.048}$ | $0.954_{\pm 0.048}$ | $0.949_{\pm 0.047}$ | $0.965_{\pm 0.032}$ |
| PSMIL ICLR2025 | $0.968_{\pm 0.053}$ | $\mathbf{0.968_{\pm 0.052}}$ | $0.942_{\pm 0.054}$ | $0.947_{\pm 0.047}$ | $0.985_{\pm 0.030}$ |
| ASMIL (Ours) | $\mathbf{0.971_{\pm 0.060}}$ | $0.968_{\pm 0.058}$ | $\mathbf{0.961_{\pm 0.025}}$ | $\mathbf{0.969_{\pm 0.037}}$ | $0.985_{\pm 0.025}$ |

Since these datasets are relatively balanced, following Du et al. (2025), we report accuracy as the primary metric. We train for 40 epochs with the Adam optimizer (Kingma & Ba, 2014) and a learning rate of 0.0005.

We compare ASMIL against six MIL methods—ABMIL (Ilse et al., 2018), DSMIL (Li et al., 2021b), TransMIL (Shao et al., 2021), DEMIL (Tang et al., 2023a), RGMIL (Du et al., 2023), and PSMIL (Du et al., 2025)—and report accuracies in Table 18. As shown, ASMIL outperforms all baselines on 4 of 5 datasets, demonstrating strong performance on non-WSI benchmarks.

# P    ATTENTION DYNAMICS OF DIFFERENT MIL METHODS ON VARIOUS DATASETS

In this section, we illustrate that the issue of attention convergence on the WSI dataset is not unique to the ABMIL and CAMELYON-16 datasets. To this end, similar to the method we describe in Figure 1, we plot the JSD of two attention distributions between two consecutive epochs.

## P.1 CAMELYON-16 DATASET

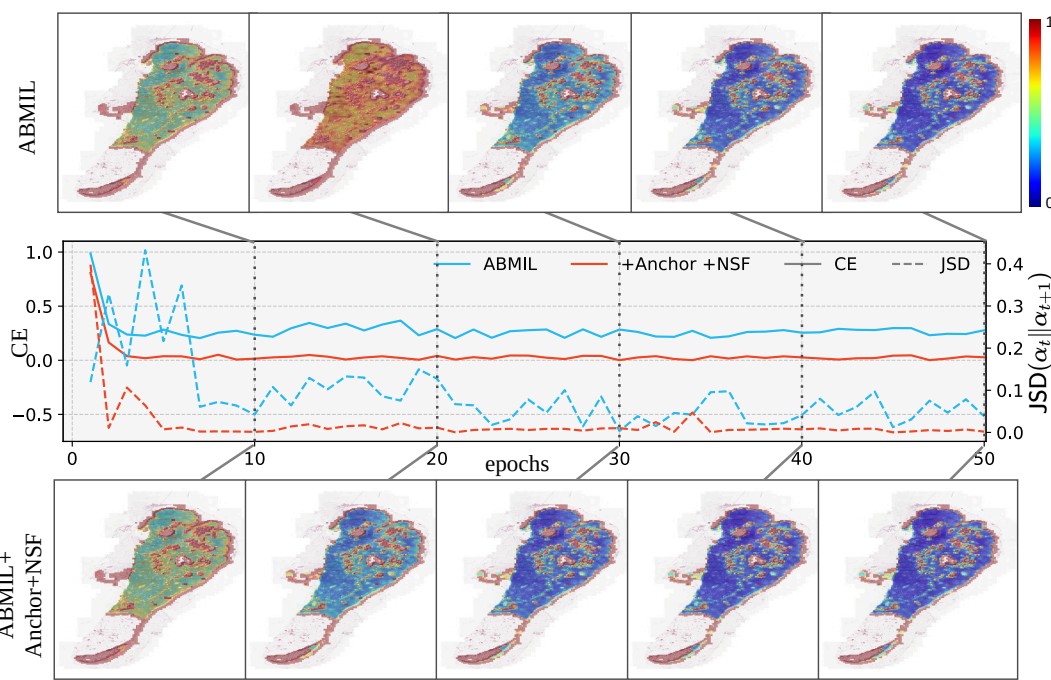

Figure 14: Visualization of attention dynamics on a normal WSI for ABMIL vs. ABMIL + anchor + NSF.

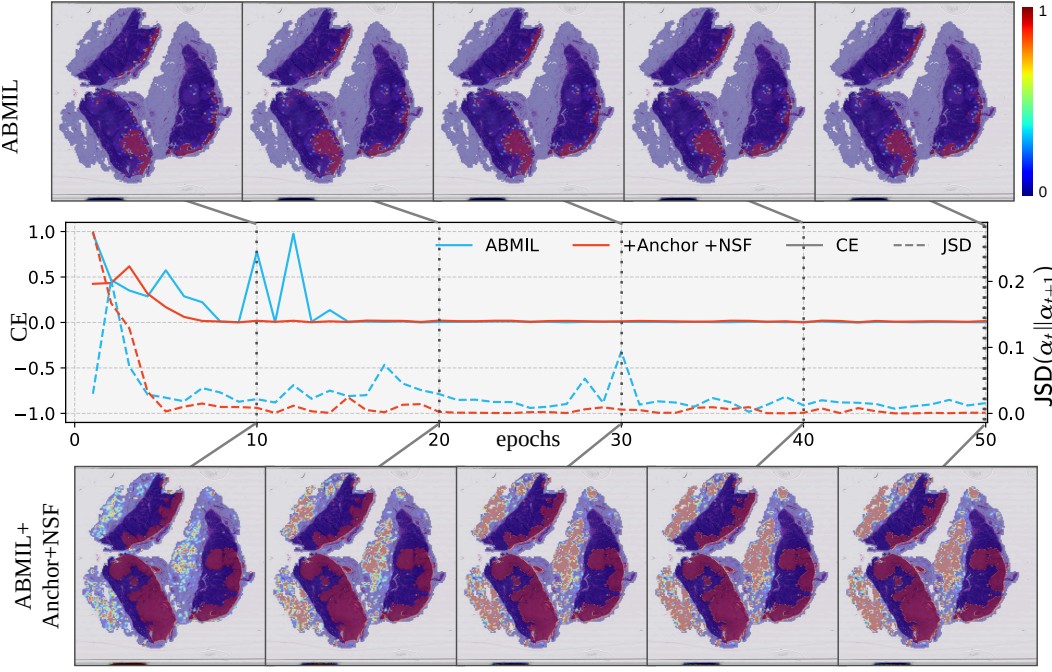

Figure 15: Visualization of attention dynamics on a tumor WSI for ABMIL vs. ABMIL + anchor + NSF.

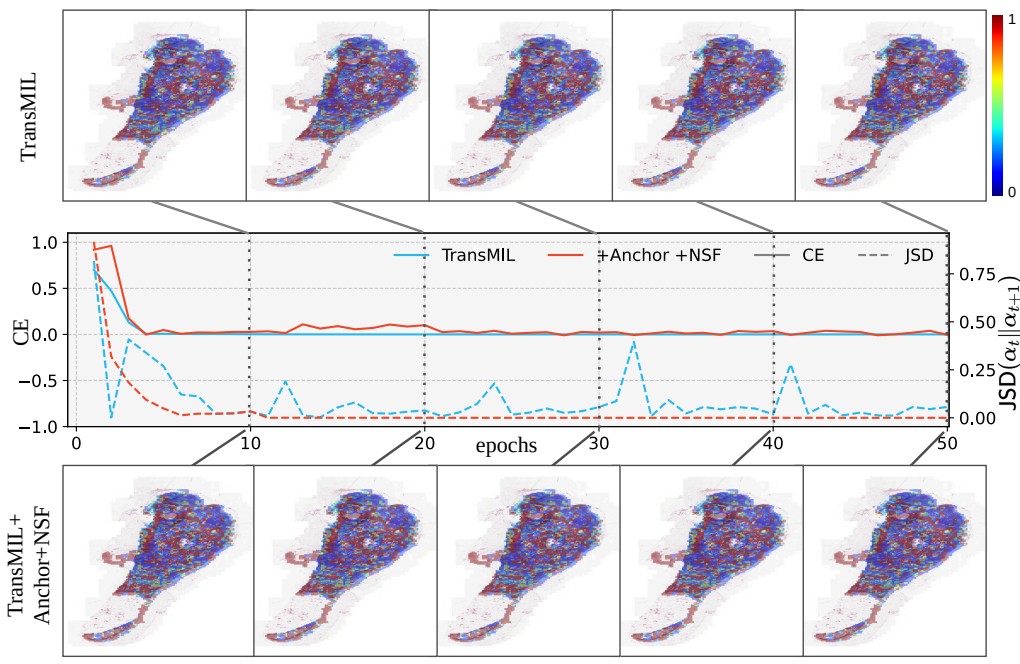

Figure 16: Visualization of attention dynamics on a normal WSI for TransMIL vs. TransMIL + anchor + NSF.

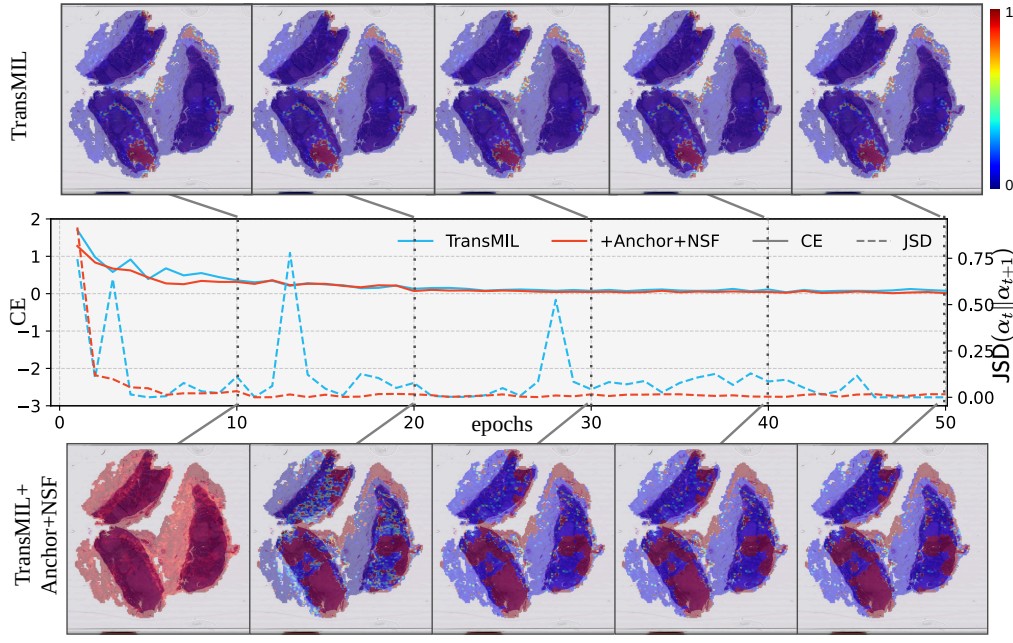

Figure 17: Visualization of attention dynamics on a tumor WSI for TransMIL vs. TransMIL + anchor + NSF.

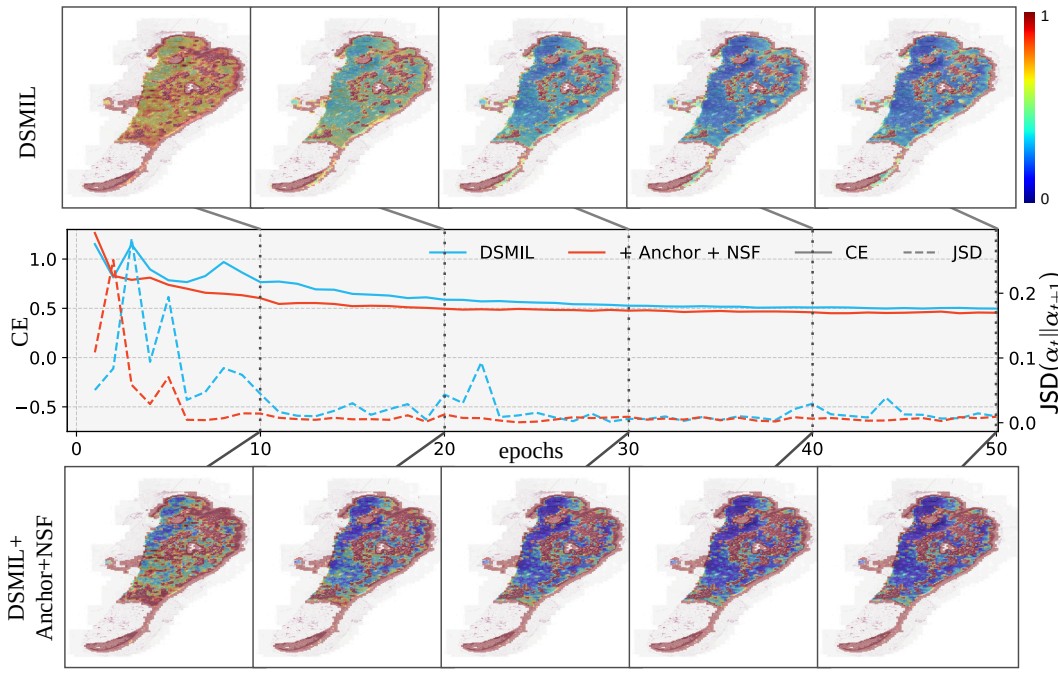

Figure 18: Visualization of attention dynamics on a normal WSI for DSMIL vs. DSMIL + anchor + NSF.

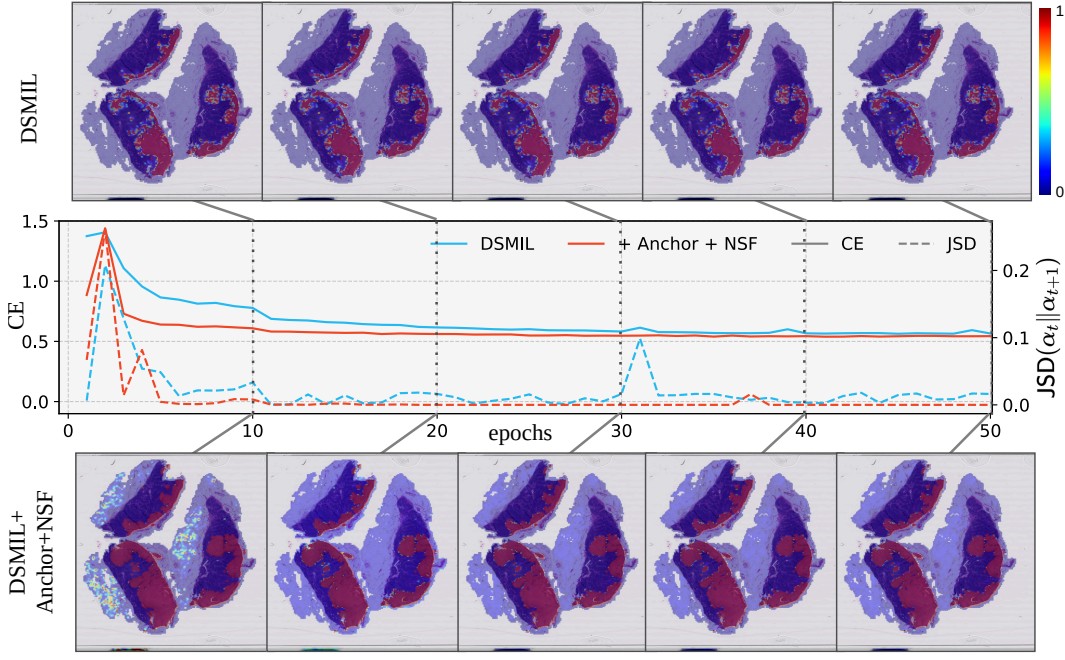

Figure 19: Visualization of attention dynamics on a tumor WSI for DSMIL vs. DSMIL + anchor + NSF.

## P.2 BRACS DATASET

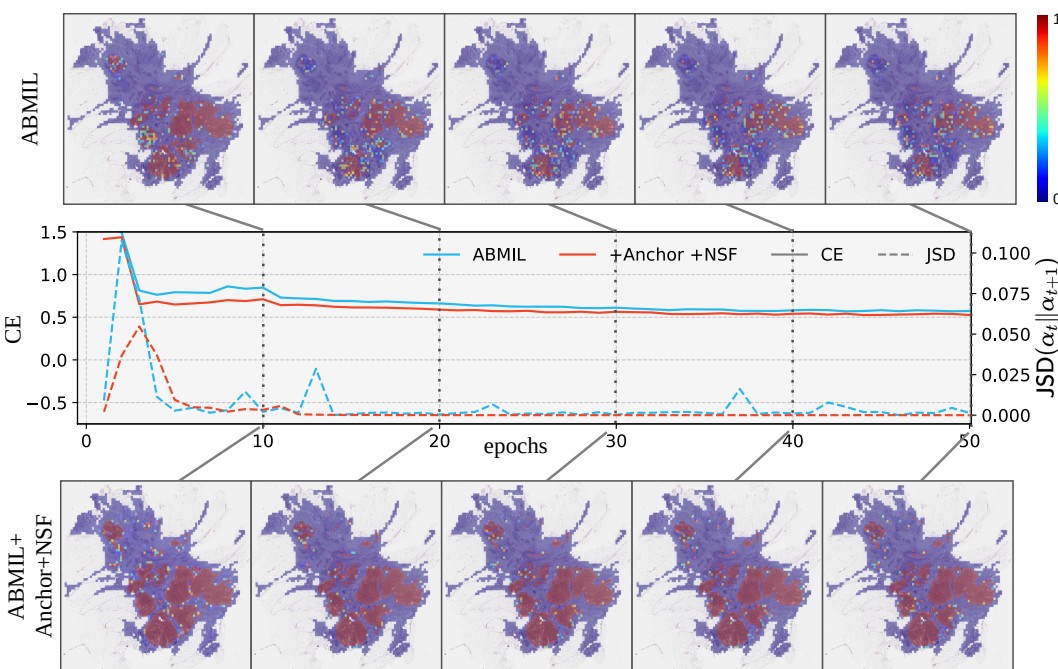

Figure 20: Visualization of attention dynamics on a normal WSI for ABMIL vs. ABMIL + anchor + NSF.

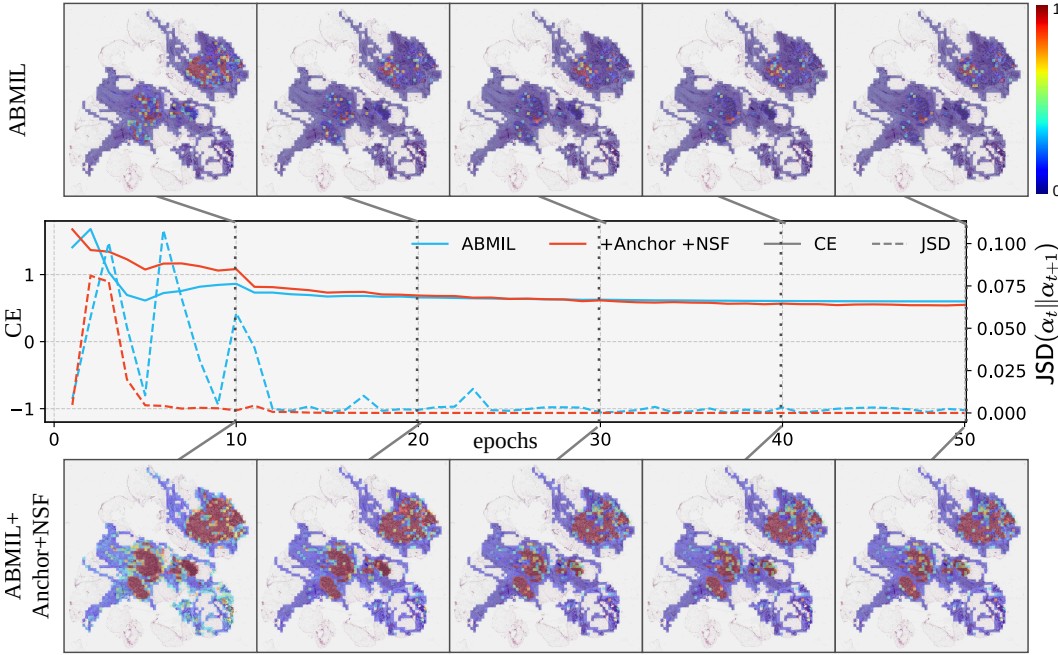

Figure 21: Visualization of attention dynamics on a tumor WSI for ABMIL vs. ABMIL + anchor + NSF.

Despite these advances, several avenues remain open for future investigation:

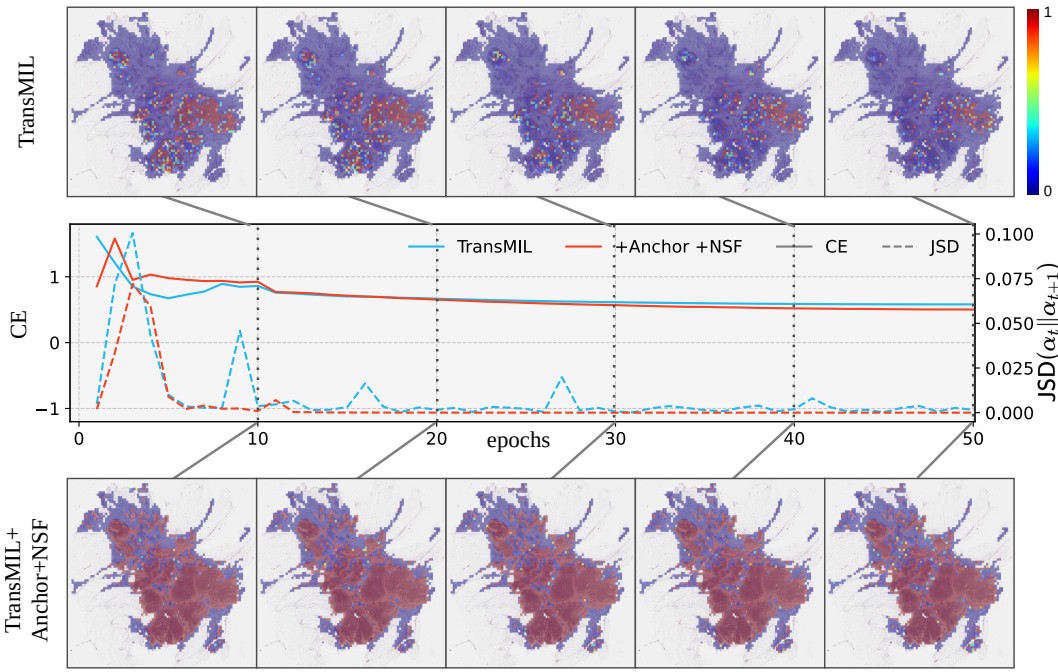

Figure 22: Visualization of attention dynamics on a tumor WSI for TransMIL vs. TransMIL + anchor + NSF.

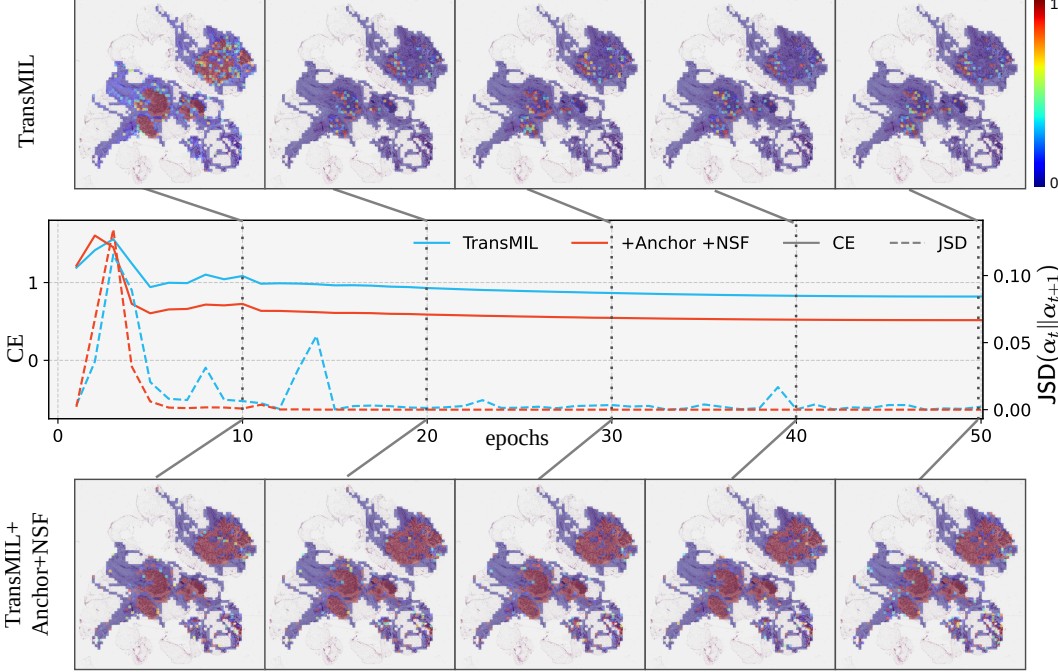

Figure 23: Visualization of attention dynamics on a tumor WSI for TransMIL vs. TransMIL + anchor + NSF.

ASMIL employs an EMA-updated anchor model to stabilize attention dynamics, but this introduces additional computational overhead. An important direction is the development of intrinsic train-

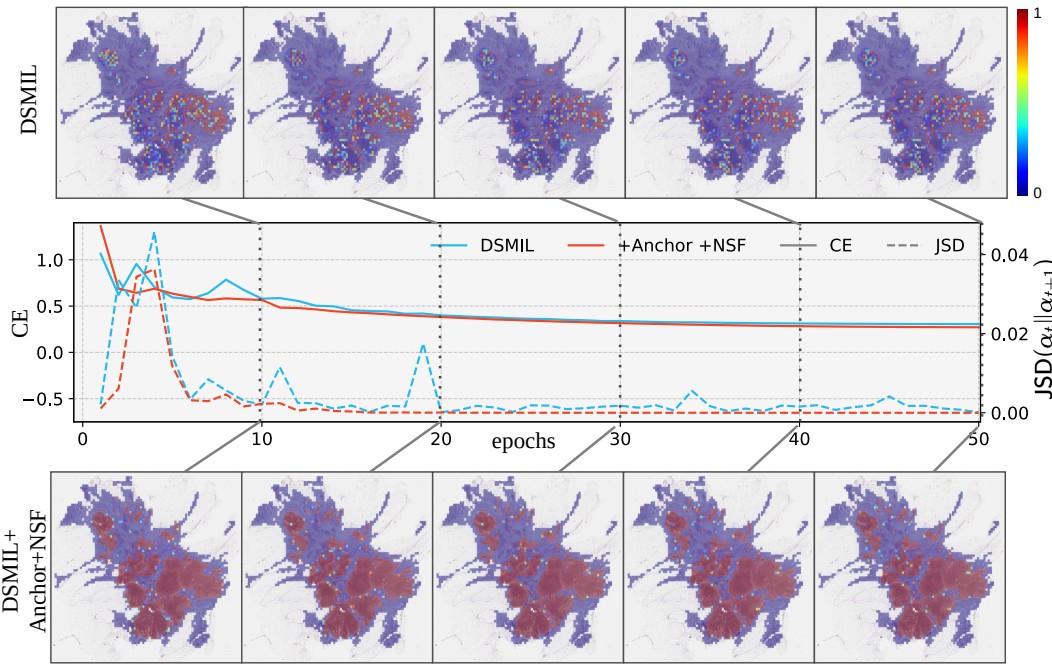

Figure 24: Visualization of attention dynamics on a tumor WSI for DSMIL vs. DSMIL + anchor + NSF.

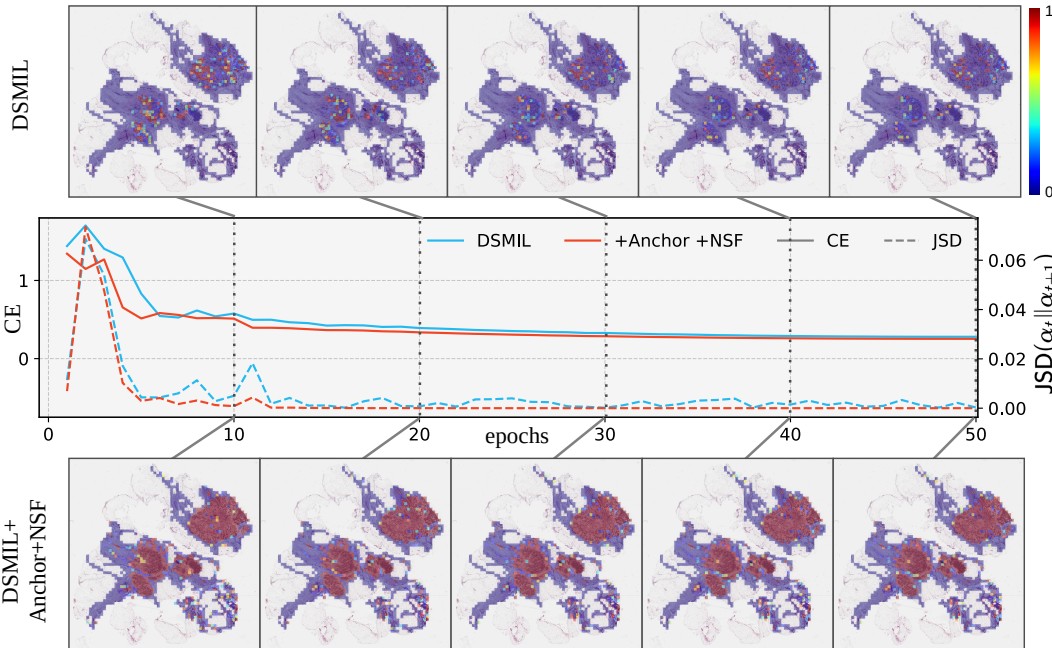

Figure 25: Visualization of attention dynamics on a tumor WSI for DSMIL vs. DSMIL + anchor + NSF.

ing strategies, such as regularization, that achieve comparable stability without auxiliary modules, thereby improving efficiency in large-scale WSI applications.

Table 19: Rate of cancerous WSIs without missed regions on the CAMELYON-16 dataset.

| Method | Clam | TransMIL | DTFD-MIL | DSMIL | CAMIL | ASMIL |
|--------|------|----------|----------|-------|-------|-------|
| Rate | 46.93% | 3.246% | 50.34% | 48.22% | 49.78% | 54.63% |

## Q LIMITATIONS AND FUTURE WORK

Meanwhile, as more advanced regularization techniques such as MIL-dropout (Zhu et al., 2025) continue to emerge, integrating them into the ASMIL framework represents a highly promising direction for future work. Such enhancements could further improve the model's generalization ability while yielding more faithful and stable attention distributions.

Furthermore, a limitation of our approach is that AS-MIL can fail by assigning low attention to tiny foci and small tumor regions (see Figure 26), particularly when large and small cancerous regions coexist within a single WSI. We fix the attention threshold at 0.5 and count a cancerous WSI as successfully localized if all regions inside the tumor annotation exceed this threshold; the success rates are reported in Table 19. ASMIL achieves the highest success rate, which we attribute to the NSF in the anchor model that mitigates over-concentrated attention. This indicates room for improvement. Nevertheless, compared with published baselines, ASMIL's attention maps consistently achieve higher Dice and FROC scores. One avenue to further enhance localization performance is to bootstrap training with a mixture of synthetic data and real WSI data. These directions are beyond the scope of this work and will be investigated in future research Wu et al. (2024); Chi et al. (2024).

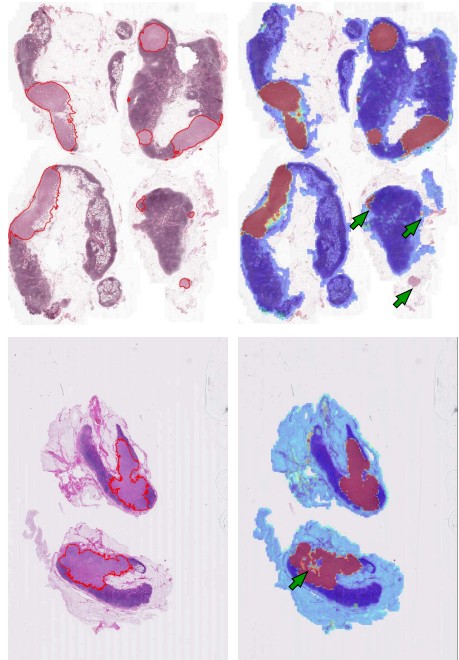

## R LLM USAGE STATEMENT

LLM used only for grammar and wording edits; no generation of ideas, methods, analyses, results, or citations. The authors reviewed all edits and accept full responsibility.

Figure 26: Left: annotated WSI. Right: attention map generated by ASMIL, which fails to assign high attention to all tumor regions, as highlighted by the green arrow.

