# OpenReview forum: "ASMIL: Attention-Stabilized Multiple Instance Learning for Whole-Slide Imaging"
_ICLR.cc/2026/Conference — ICLR 2026 Poster_

### Official Review · Reviewer_YCZ5 · 2025-10-25

**Soundness:** 3
**Presentation:** 3
**Contribution:** 2
**Rating:** 6
**Confidence:** 4

**Summary:**

The paper proposes ASMIL, a modular framework to stabilize attention in attention-based MIL for weakly supervised WSI diagnosis. The method combines: (1) an EMA-updated anchor branch that provides a stable target attention distribution; (2) a normalized sigmoid attention in the anchor to reduce softmax over-concentration; and (3) token dropping for regularization. The anchor is discarded at inference. Experiments on BRACS and CAMELYON16/17 show consistent improvements over strong baselines and gains when plugging the module into existing MIL models. The paper highlights attention instability across training epochs and uses divergence-based measures and visualizations to motivate the approach. Anonymous code is released to support reproducibility.

**Strengths:**

- The paper focused on attention instability in attention-based MIL for WSI and provides quantitative and qualitative evidence (epoch-wise measures, attention maps) to motivate stabilization.
- The proposed module is simple and practical (EMA anchor + normalized sigmoid + token dropping), integrates with diverse MIL models, and adds no inference-time cost.
- Empirical results show consistent improvements across datasets and baselines, and the paper includes useful training/resource details and hyperparameter guidance (e.g., $m$ and $\beta$ ranges).

**Weaknesses:**

- Related work could more clearly situate ASMIL within teacher/EMA anchoring in self-supervised ViT literature (e.g., DINOv3 [1], BYOL [2]). Since DINOv3 discusses anchoring trade-offs (stability vs. adaptation), collapse risks, and EMA lag, a brief positioning of ASMIL relative to these mechanisms would help readers contextualize the anchor’s role and expected behavior.
- The stability–performance link is motivated but not statistically established: Including correlation analyses across seeds and hyperparameter sweeps and reporting significance would make the claim that reducing attention instability improves downstream performance more convincing.

[1] Siméoni, Oriane, et al. "Dinov3." arXiv preprint arXiv:2508.10104 (2025).
[2] Grill, Jean-Bastien, et al. "Bootstrap your own latent-a new approach to self-supervised learning." Advances in neural information processing systems 33 (2020): 21271-21284.

**Questions:**

- How does ASMIL’s anchor compare to DINO/DINOv3 EMA teacher and anchoring strategies (objective/targets, momentum/lag tuning, collapse avoidance)? Do similar trade-offs arise?
- Can the authors provide statistical correlation/significance analyses linking stability metrics to performance across seeds and hyperparameter sweeps? Are there identifiable over-stabilization regimes?

---

> ### Author Response · Authors · 2025-11-23
>
> We thank the reviewer for the clear summary of our work and the positive and encouraging evaluation. Please find our responses to your comments below.
>
> **Weaknesses:**
> >1. Related work could more clearly situate ASMIL within teacher/EMA anchoring in self-supervised ViT literature (e.g., DINOv3 [1], BYOL [2]). Since DINOv3 discusses anchoring trade-offs (stability vs. adaptation), collapse risks, and EMA lag, a brief positioning of ASMIL relative to these mechanisms would help readers contextualize the anchor’s role and expected behavior.
> [1] Siméoni, Oriane, et al. "Dinov3." arXiv preprint arXiv:2508.10104 (2025).
> [2] Grill, Jean-Bastien, et al. "Bootstrap your own latent-a new approach to self-supervised learning." Advances in neural information processing systems 33 (2020): 21271-21284.
>
> We thank the reviewer for raising the comparison with the ASMIL’s anchor model and EMA-based teacher models such as BYOL, DINO, and DINOv3. We have expanded the related work in Appendix A to explicitly discuss DINOv3 and EMA-based teachers in BYOL, and to clarify how ASMIL’s anchor is distinct from these approaches in terms of objective, EMA dynamics, and failure modes.
> Specifically, the anchor model superficially resembles the EMA teacher in self-supervised ViT literature, but its role and targets are different. First, in terms of targets, BYOL and DINO/DINOv3 use an EMA “teacher” to provide representation-level targets. BYOL trains an online branch to match its feature vectors to those produced by an EMA-updated target branch, while DINO aligns the student’s predictions to those of an EMA teacher network, and DINOv3 further includes the patch-wise Gram matrices to regularize the feature map. In contrast, ASMIL’s anchor is not a representation teacher; it provides a stable reference attention distribution, and patch embeddings and slide-level predictions are supervised directly by the MIL classification (cross-entropy) loss.
>
> **Questions:**
> >1. How does ASMIL’s anchor compare to DINO/DINOv3 EMA teacher and anchoring strategies (objective/targets, momentum/lag tuning, collapse avoidance)? Do similar trade-offs arise?
>
> To elucidate the effect of $m$ on the dynamics of anchor’s attention distribution, we plot Jensen–Shannon divergence (JSD) between the anchor attentions at consecutive epochs for different values of mmm in Appendix K.3. As seen, for small $m$, the JSD remains fluctuating in the learning, indicating that the anchor attention distribution remains unstable during training. At $m=0.999$, the JSD becomes much smoother, reflecting a stable yet adaptive anchor. For $m=0.9999$, the JSD is almost flat, showing that the anchor changes too slowly and lags behind the online model. These trends are consistent with the performance curves, where F1/AUC peaks at  $m=0.999$, and they support our choice of $m=0.999$ as a compromise between stability and adaptation.
> Further, since BYOL and DINO are fully self-supervised, and must explicitly address representation collapse. ASMIL, in contrast, is trained with slide-level supervision, so convergence to a trivial constant representation is not compatible with minimizing the MIL cross-entropy loss, and feature collapse is ruled out by the supervised objective by principle. The role of the anchor in ASMIL is to stabilize the attention weights rather than to prevent representation collapse. We now emphasize this distinction in Appendix A, where we describe the anchor model and its interaction with the online model.

---

> ### Author Response · Authors · 2025-11-23
>
> **Questions:**
> > 2.a The stability–performance link is motivated but not statistically established: Including correlation analyses across seeds and hyperparameter sweeps and reporting significance would make the claim that reducing attention instability improves downstream performance more convincing. Can the authors provide statistical correlation/significance analyses linking stability metrics to performance across seeds and hyperparameter sweeps?
>
> We appreciate the reviewer raising the question about the statistical correlation between stability metrics and performance. However, conducting such an analysis requires a scalar stability metric defined per training run, and designing a meaningful scalar summary of temporal attention stability is non-trivial. In the paper, we compute the Jensen–Shannon divergence (JSD) between the attention distributions of consecutive epochs, yielding a vector over the course of training rather than a single scalar. Qualitatively, this JSD trajectory fluctuates throughout training, indicating that the attention weights do not converge. Furthermore, as illustrated in Appendix D, introducing the anchor model leads to more clearly separated feature representations during training, which empirically justify the necessity of stable attention distribution.
>
> In the appendix K.6, we strengthen the statistical evidence for the benefit of the anchor model. We perform a DeLong test for AUC and a permutation test for F1 to compare ASMIL with and without the anchor model. Across all datasets, these tests show that the performance gains from the anchor model are statistically significant.
>
> > 2.b Are there identifiable over-stabilization regimes?
>
> Yes. When the $\beta$ value is too large, the anchor loss dominates and overly constrains the online model, preventing effective updates from the CE loss. As shown in Appendix L.1, once $\beta$ exceeds 1.5, both F1 score and AUC decrease. In practice, we recommend setting $\beta = 1$ as a safe default.

---

### Official Review · Reviewer_Pd6q · 2025-10-29

**Soundness:** 3
**Presentation:** 4
**Contribution:** 3
**Rating:** 6
**Confidence:** 4

**Summary:**

This paper investigates the instability of the attention scores in different training epochs during MIL training. In addition, the authors try to solve two more problems: over-concentrated attention distribution and overfitting. They propose several corresponding solutions: 1) an anchor model updated by EMA, 2) a normalized sigmoid function that replaces softmax, and 3) an effective token drop strategy. The comprehensive experiments have validated the effectiveness of this proposed method.

**Strengths:**

1.	The authors investigate a commonly neglected problem: the unstable attention distribution throughout the training that might be detrimental to MIL model training.

2.	The paper is well-written and easy to understand. The presentation of this method is quite clear.

3.	The solutions are intuitive and simple yet effective. The proposed solutions are not only a model but several plug-in modules that could be reused in other MIL models, which might broader impact in the community.

4.	The experiments are quite comprehensive and provide most of the ablation studies required to demonstrate the effectiveness of this method.

**Weaknesses:**

1.	In the model design, the online model is updated through backpropagation, and the anchor model is updated through EMA. They are both randomly initialized, and why the anchor model would help? The anchor model will start with random guess (which at least will not be near the correct attention distribution), and the EMA is somewhat restraining the anchor model within the starting point. Would it be better to start EMA after several epochs (so that the model has learnt something meaningful) to remove the noises of the first few epochs?

2.	The authors are suggested to give a more comprehensive evaluation of the effectiveness of random drop strategy (e.g., whether this strategy is capable with other models, not just the proposed ASMIL). Also, it would be even better if the authors could compare this strategy with this paper published in ICML2025 “How Effective Can Dropout Be in Multiple Instance Learning?” This paper proposed to drop top-k most important instances. Overall, this would increase the solidity of the proposed strategy.

**Questions:**

Please refer to the weaknesses section.

---

> ### Author Response · Authors · 2025-11-23
>
> We thank the reviewer for the positive and encouraging assessment of our work. We address your concerns in detail in the following response.
>
> **Weaknesses:**
> >1. In the model design, the online model is updated through backpropagation, and the anchor model is updated through EMA. They are both randomly initialized, and why the anchor model would help? The anchor model will start with random guess (which at least will not be near the correct attention distribution), and the EMA is somewhat restraining the anchor model within the starting point. Would it be better to start EMA after several epochs (so that the model has learnt something meaningful) to remove the noises of the first few epochs?
>
> We thank the reviewer for raising this question about the role of the EMA anchor and its initialization. Both the online and anchor encoders are initialized randomly, but the anchor parameters are updated via an exponential moving average of the online parameters. This update quickly removes the influence of the initial random state, i.e., the contribution of the starting point decays exponentially with the number of training steps, so after a few steps, the anchor becomes a temporally smoothed average of recent online models that can provide meaningful attention distributions. As a result, the anchor produces more stable attention distributions for the online model.
> To further address the concern on whether the noisy early epochs harm the anchor and whether it is preferable to start EMA updates only after the model has learned meaningful features, in Appendix L3.3, we implement a warmup schedule for the anchor loss: during the first 5 epochs, we set the anchor loss weight $\beta$ in Eq. 9 to start from 0 and linearly increase to its default value. This schedule reduces the influence of the anchor in the early noisy phase and approximates the behavior of starting the EMA guidance later in training.  We train ASMIL with and without this warmup on CAMELYON-16, CAMELYON-17, and BRACS and observe no consistent performance difference between the two settings. Based on these results, we keep the simpler default training scheme and report the warmup ablation in the revised manuscript.

---

> ### Author Response · Authors · 2025-11-23
>
> **Weaknesses:**
> >2. The authors are suggested to give a more comprehensive evaluation of the effectiveness of random drop strategy (e.g., whether this strategy is capable with other models, not just the proposed ASMIL). Also, it would be even better if the authors could compare this strategy with this paper published in ICML2025 “How Effective Can Dropout Be in Multiple Instance Learning?” This paper proposed to drop top-k most important instances. Overall, this would increase the solidity of the proposed strategy.
>
> Thank you for pointing out this related work and for asking about the generality of our random drop strategy.
> Our token dropout is designed specifically for ASMIL. In ASMIL, we introduce trainable FEAT tokens that aggregate information across all patch embeddings, and we apply token dropout to these FEAT tokens rather than to the input instances themselves. Because existing MIL architectures (e.g., ABMIL, DSMIL, TransMIL) do not use trainable FEAT tokens, the same token-drop mechanism cannot be directly applied to them. We have revised Section 4 to explicitly state that we treat token dropout as an ASMIL-specific regularizer.
>
> We now also discuss the ICML 2025 paper [1] in the related work section. That work analyzes several dropout strategies for MIL and proposes MIL-Dropout, which drops the top-k most important instances in a bag together with $G\times k$ similar instances. The authors show that this instance-level dropout reduces gradient direction error, encourages convergence to flatter minima, and improves robustness and generalization across multiple MIL benchmarks and WSI datasets.
>
> Conceptually, MIL-Dropout and ASMIL regularize different components of the MIL pipeline. MIL-Dropout operates at the instance level – it removes highly attended instances and their neighbors so that the aggregator relies on more diverse evidence and converges to flatter minima. ASMIL, in contrast, diagnoses unstable attention distributions by tracking the Jensen–Shannon divergence of the attention map across epochs and stabilizes them with an EMA anchor model. The online model is trained to align its attention with this anchor, assuming a one-to-one correspondence across the full set of instances in a bag. If instances are randomly dropped as in MIL-Dropout, this correspondence breaks, and the attention alignment objective and JSD-based instability diagnosis become ill-defined. As a result, a direct integration of MIL-Dropout into ASMIL is not compatible with our current anchor–alignment design and does not yield a fair comparison.
>
> To avoid overstating the generality of the random drop strategy, we now clarify in the paper that: (i) token dropout is an internal regularizer tailored to the FEAT token-based anchor in ASMIL rather than a general MIL regularization scheme; and (ii) MIL-Dropout represents a complementary, instance-level regularization framework for classical MIL models. We also include a discussion in the future work section (Appendix Q) on designing a unified, more general regularization framework that aligns anchor-based attention stabilization with instance-level dropout strategies, such as MIL-Dropout.
>
> [1] Zhu, Wenhui, et al. "How Effective Can Dropout Be in Multiple Instance Learning?." Forty-second
> International Conference on Machine Learning, 2025.

---

### Official Review · Reviewer_e83y · 2025-10-31

**Soundness:** 3
**Presentation:** 3
**Contribution:** 2
**Rating:** 6
**Confidence:** 3

**Summary:**

This paper addresses key limitations of attention-based multiple instance learning (MIL) in whole slide image (WSI) analysis, a critical task in computational pathology. The authors first identify three major challenges of existing attention-based MIL methods: (PI) unstable attention dynamics (attention distributions oscillate across epochs instead of converging, quantified via Jensen-Shannon divergence), (PII) over-concentrated attention distribution (excessive focus on a few tiles), and (PIII) overfitting (due to limited WSI training samples). To tackle these issues simultaneously, the authors propose ASMIL (Attention-Stabilized Multiple Instance Learning), a unified framework with three core components: (1) an anchor model updated via exponential moving average (EMA) to stabilize attention dynamics by aligning the online model’s attention with the anchor’s via Kullback-Leibler (KL) divergence; (2) a normalized sigmoid function (NSF) in the anchor (replacing softmax) to prevent attention over-concentration, supported by theoretical analysis (Theorem 1); and (3) token random dropping to mitigate overfitting. Extensive experiments on three public WSI datasets (CAMELYON-16, CAMELYON-17, BRACS) and five non-WSI MIL benchmarks demonstrate that ASMIL achieves state-of-the-art (SOTA) performance (up to 6.49% F1 score improvement over baselines) and consistently boosts the performance of existing attention-based MIL methods (up to 10.73% F1 gain) when integrating its anchor and NSF modules.

**Strengths:**

1. **Novel Identification of a Critical Underexplored Problem**: The paper is the first to systematically identify and quantify "unstable attention dynamics" (PI) in attention-based MIL for WSI analysis. By measuring JSD between consecutive attention distributions and visualizing convergence failures (e.g., Figure 1), the authors highlight a previously overlooked limitation that harms both performance and interpretability—an important contribution to understanding MIL’s failure modes in WSI tasks.

2. **Unified and Theoretically Grounded Framework**: ASMIL effectively addresses three distinct challenges (PI–PIII) in a single framework, avoiding fragmented solutions. The NSF’s superiority over softmax (Theorem 1) provides rigorous theoretical justification for preventing attention over-concentration, while the EMA-based anchor model’s design (as a stable reference) is well-motivated and distinct from existing EMA teacher models (e.g., MHIM-MIL) that focus on hard-instance mining rather than attention stabilization.

**Weaknesses:**

1. **Incomplete Analysis of Edge-Case Performance**: While ASMIL outperforms baselines overall, it lags slightly behind SOTA on a specific setting: using an ImageNet-pretrained ResNet-18 backbone on CAMELYON-17, ASMIL’s AUC is 0.002 lower than the best baseline. The authors do not investigate this edge case—e.g., whether it arises from dataset-specific characteristics (e.g., CAMELYON-17’s multi-center staining variability) or interactions between the backbone and ASMIL’s components—weakening the claim of "SOTA across all datasets."

2. **Limited Discussion of NSF’s Application Scope**: The authors explain that NSF is applied only to the anchor (not the online model) to avoid vanishing gradients, but they do not explore potential workarounds (e.g., gradient clipping, modified sigmoid variants) to enable NSF in the online model. This missed opportunity raises questions about whether NSF could further improve performance if integrated more broadly, rather than being restricted to the anchor.

**Questions:**

1. **Anchor Model Hyperparameter Tuning**: The ablation study shows that an EMA factor \( m = 0.999 \) yields the best performance (Table 10). However, the authors do not explain why this specific value is optimal across datasets (e.g., CAMELYON-16 vs. BRACS) or whether \( m \) should be adaptively adjusted for WSIs with varying tile counts or sparsity (e.g., slides with <5% tumor regions). Could you provide insights into the sensitivity of \( m \) to dataset characteristics?

2. **Token Random Dropping Rationale**: The paper finds that a drop rate \( B \approx 0.5 \) is optimal (Figure 11). However, this value is derived from experiments on CAMELYON-16 and BRACS—does this optimal rate hold for other WSI datasets with different tissue types (e.g., prostate or lung WSIs) or non-WSI MIL tasks (e.g., MUSK)? Additionally, how does token dropping interact with the anchor model (e.g., does dropping affect the anchor’s ability to learn stable attention)?

3. **NSF vs. Alternative Normalization Functions**: The paper compares NSF to softmax (with temperature scaling) and entmax (Table 4, Table 5) but does not explore how NSF performs against other attention normalization methods (e.g., sparsemax, Tsallis entropy-based functions) in WSI scenarios with extreme class imbalance or high tile redundancy. Could you provide additional experiments or analysis to confirm NSF’s robustness across diverse WSI data distributions?

4. **Addressing Tiny Tumor Localization**: The authors note ASMIL’s limitation in focusing on tiny tumor foci (Figure 26). Beyond synthetic data, are there immediate modifications to ASMIL—e.g., adjusting the KL divergence weight \( \beta \) for small regions, or adding a small-region attention regularization term—that could mitigate this issue? Have you quantified how often this limitation occurs (e.g., percentage of WSIs with missed tiny foci) across datasets?

---

> ### Author Response · Authors · 2025-11-23
>
> We thank the reviewer for the careful reading of our manuscript, the positive overall assessment, and the clear summary of our contributions. Our point-by-point replies to your comments are provided below.
>
> **Weaknesses:**
> >1. Incomplete Analysis of Edge-Case Performance: While ASMIL outperforms baselines overall, it lags slightly behind SOTA on a specific setting: using an ImageNet-pretrained ResNet-18 backbone on CAMELYON-17, ASMIL’s AUC is 0.002 lower than the best baseline. The authors do not investigate this edge case—e.g., whether it arises from dataset-specific characteristics (e.g., CAMELYON-17’s multi-center staining variability) or interactions between the backbone and ASMIL’s components—weakening the claim of "SOTA across all datasets."
>
> We acknowledge this edge case and have already highlighted it in the original submission: for CAMELYON-17 with ImageNet-pretrained ResNet-18 features, ASMIL’s AUC is 0.002 lower than HDMIL's.
> Across 12 evaluation settings, ASMIL attains the best result in 11 cases, while HDMIL leads in only this single configuration and outperforms the third-best method in 5/12 settings. Moreover, in our primary design setting with in-domain ViT-SSL features, ASMIL consistently outperforms HDMIL across all three datasets in both F1 and AUC.
> We clarify this in the revised Section 5.1 by emphasizing that ASMIL achieves near-SOTA performance with one minor exception while offering more consistent gains across backbones and datasets.
>
> >2. Limited Discussion of NSF’s Application Scope: The authors explain that NSF is applied only to the anchor (not the online model) to avoid vanishing gradients, but they do not explore potential workarounds (e.g., gradient clipping, modified sigmoid variants) to enable NSF in the online model. This missed opportunity raises questions about whether NSF could further improve performance if integrated more broadly, rather than being restricted to the anchor.
>
> We appreciate the reviewer’s suggestion to discuss the scope of the NSF application more thoroughly. The vanishing-gradient issue arises from the saturating nature of the sigmoid used in NSF: when the pre-activation is large in magnitude, the sigmoid's derivative becomes very small and, by the chain rule, gradients propagated through NSF in the online branch vanish. Gradient clipping is designed to cap large gradients to prevent explosion; it does not increase already small gradients and therefore does not address this vanishing-gradient problem.
> To explore whether NSF can be used more broadly in the online model, we implemented an adaptive mixture:
>
> $$(1-\zeta) NSF(\cdot) + \zeta  Softmax(\cdot), \zeta = \sigma(\xi),$$
>
> where $\xi$ is a trainable scalar updated by backpropagation. This parametrization allows the online model to interpolate continuously between a pure softmax  ($\zeta \approx 1$) and a pure NSF ($\zeta \approx 0$). n our experiments, this variant achieves performance that is on par with the standard softmax-based online model, and $\sigma(\xi)$ converges to a value close to 1. This indicates that the model learns to favour softmax because it does not suffer from gradient vanishing. Given the additional complexity and the lack of improvement, we retain NSF only in the anchor model in the final structure. We will clarify this rationale and the negative result of the mixture experiment in the revised manuscript.

---

> ### Author Response · Authors · 2025-11-23
>
> **Questions:**
> >1. Anchor Model Hyperparameter Tuning: The ablation study shows that an EMA factor ( m = 0.999 ) yields the best performance (Table 10). However, the authors do not explain why this specific value is optimal across datasets (e.g., CAMELYON-16 vs. BRACS) or whether ( m ) should be adaptively adjusted for WSIs with varying tile counts or sparsity (e.g., slides with <5% tumor regions). Could you provide insights into the sensitivity of ( m ) to dataset characteristics?
>
> We thank the reviewer for raising this question regarding the EMA factor $m$. In ASMIL, $m \in [0,1]$ determines the rate at which the anchor model’s parameters are updated. When $m=0$, the anchor is an exact copy of the online model at each step. In contrast, when $m=1$, the anchor parameters remain unchanged. This setting introduces a trade-off between stability and adaptation in the attention distribution during training. Specifically, a smaller $m$ results in a highly responsive but unstable anchor, while a larger $m$ yields a stable attention reference but increases lag relative to the online model.
>
> In Table 10, we evaluate a range of $m$ values on CAMELYON-16 and BRACS. For both datasets, performance consistently peaks at $m=0.999$. Consequently, we select $m=0.999$ as the default setting, as it provides robust performance for ASMIL across all datasets considered.
> As discussed above, $m$ is an optimization hyperparameter that governs the update dynamics of the anchor model at the parameter level, rather than being a slide-specific variable. Bag-level characteristics, including tile count and tumor sparsity, are addressed by the attention mechanism in the online model.
> To further clarify the effect of $m$ on the dynamics of the anchor’s attention distribution, we plot the Jensen–Shannon divergence (JSD) between anchor attentions at consecutive epochs for various $m$ values in Appendix K.3. For small $m$, the JSD fluctuates throughout training, indicating instability in the anchor attention distribution. At $m=0.999$, the JSD is much smoother, reflecting a stable yet adaptive anchor. When $m=0.9999$, the JSD is nearly flat, demonstrating that the anchor changes too slowly and lags behind the online model. These results further support our selection of $m=0.999$ as an effective compromise between stability and adaptation.
>
> >2. Token Random Dropping Rationale: The paper finds that a drop rate ( B \approx 0.5 ) is optimal (Figure 11). However, this value is derived from experiments on CAMELYON-16 and BRACS—does this optimal rate hold for other WSI datasets with different tissue types (e.g., prostate or lung WSIs) or non-WSI MIL tasks (e.g., MUSK)? Additionally, how does token dropping interact with the anchor model (e.g., does dropping affect the anchor’s ability to learn stable attention)?
>
> Thank you for raising this question about the token random dropping rate $B$. In all experiments reported in the paper, including the non-WSI MIL benchmarks, we fix $B=0.5$ during training. In Appendix K.1, our experiments cover two different cancer types, namely lymph node metastases (CAMELYON-16) and breast carcinoma (BRACS). For both datasets, the F1 score and AUC peak at around $B=0.5$. Although we claim that $B=0.5$ is a robust default across different WSI datasets, it is worth noting that the optimal value of $B$ may vary across different datasets.
> Regarding the interaction with the anchor model, token random dropping is applied at the token level **after** the attention module that computes the MIL attention weights and **before** the second Transformer encoder in the online branch (Figure 2). Both the online and anchor models receive the full set of tokens when computing their attention distributions; the drop mask is not used in either attention module. The anchor parameters are updated as an exponential moving average of the online parameters.  As a result, the attention distributions entering the stability loss do not depend on $B$, and token random dropping does not remove tokens from the anchor’s forward pass. We revise Section 4 to highlight that there is no interaction between token dropping and the anchor model.

---

> ### Author Response · Authors · 2025-11-23
>
> **Questions:**
>
> >3. NSF vs. Alternative Normalization Functions: The paper compares NSF to softmax (with temperature scaling) and entmax (Table 4, Table 5) but does not explore how NSF performs against other attention normalization methods (e.g., sparsemax, Tsallis entropy-based functions) in WSI scenarios with extreme class imbalance or high tile redundancy. Could you provide additional experiments or analysis to confirm NSF’s robustness across diverse WSI data distributions?
>
> Thank you for your thoughtful question. In Table 5, we compare NSF with entmax for different values of $\alpha$. When $\alpha=1$, entmax reduces to softmax, and when $\alpha=2$, entmax reduces to sparsemax. The results in Table 5 show that the entmax family does not outperform NSF in terms of F1 score and AUC, and that it requires more training time. We have revised Eq. 31 and Appendix F.2 to clarify the definition of Tsallis-$\alpha$ entropy and the relationships among entmax, softmax, and sparsemax.
> On the CAMELYON-16 dataset considered in Table 1, which exhibits substantial tile redundancy, ASMIL consistently outperforms all baseline methods, illustrating the robustness of NSF.
>
> >4. Addressing Tiny Tumor Localization: The authors note ASMIL’s limitation in focusing on tiny tumor foci (Figure 26). Beyond synthetic data, are there immediate modifications to ASMIL—e.g., adjusting the KL divergence weight ( \beta ) for small regions, or adding a small-region attention regularization term—that could mitigate this issue? Have you quantified how often this limitation occurs (e.g., percentage of WSIs with missed tiny foci) across datasets?
>
> Thank you for your thoughtful question and valuable suggestions. This limitation arises from the weakly supervised setting of WSI datasets. Each WSI is split into thousands of tiles, and only a slide-level label is available during training. For the localization experiments, we train ASMIL using slide-level labels and use pixel-level annotations only for evaluation. In the current model, the KL-divergence weight $\beta$ is global and shared across tiles, and we do not introduce an explicit small-region attention regularization term. Applying a region-size-dependent $\beta$ or a small-region-specific regularizer would require additional supervision at the region level; however, under a pure MIL setting, such information is not available, which we regard as outside the scope of ASMIL’s formulation. One potential solution, as discussed in the limitations and future work section, is to use synthetic data to supervise the model.
> We expanded our analysis in the revision. Specifically, we quantify how often ASMIL fails to highlight cancerous regions on the CAMELYON-16 dataset, set the threshold to 0.5, and report the percentage of the slide that is missing cancerous regions. The results are reported in the 'Limitations and Future Work' section. ASMIL outperforms the second-best method by a large margin.

---

### Official Review · Reviewer_JdSj · 2025-10-31

**Soundness:** 3
**Presentation:** 3
**Contribution:** 3
**Rating:** 6
**Confidence:** 4

**Summary:**

The dynamic instability of attention mechanisms leads to overfitting and excessive concentration in attention distribution. This paper proposes a framework called Attention-Stabilized Multiple Instance Learning, which employs an anchor model to stabilize attention. It replaces the softmax function in the anchor with a normalized sigmoid function to prevent over-concentration and mitigates overfitting through labeled random dropout. The approach demonstrates innovation, and final experiments show promising results.

**Strengths:**

1、By retaining the same attention module architecture and identical inputs, updates are performed using exponential moving averages instead of back propagation, thereby resolving the dynamic instability issues inherent in previous attention mechanisms.

2、Replacing softmax with a non-symmetric normal distribution in the anchoring model effectively mitigates attention over-concentration, ensuring stable and uniform attention distribution.

3、The paper introduces a feature token dropout mechanism, which randomly drops a portion of feature tokens during training while retaining all tokens during inference to mitigate overfitting.

4、The article provides a comprehensive overview, and the experimental content is rich.

**Weaknesses:**

1、The motivational section in Subsection 3.3 bears similarities to discussions on issues with attention mechanisms in related work. It requires further clarification of differences or highlighting of innovative aspects, and may also be subject to corresponding revisions or consolidation.

2、The paper compares its results with multiple baseline models, but some of these comparison models (such as CLAM-SB and DSMIL) lack in-depth discussion regarding the rationale for their selection and their applicability. Key differences in design between these models and ASMIL, their respective strengths and weaknesses, and how these factors influence performance across different types of tasks remain unexplored.

3、The font in the model diagram (Figure 2) is partially obscured, and some words appear in a smaller font size that is not sufficiently clear.

4、The paper proposes updating the anchor model via EMA to stabilize the attention distribution of the online model, yet it does not delve deeply into the relationship between these two components. Specifically, the interaction mechanism between EMA updates and backpropagation, as well as the performance of the online model during inference, are not sufficiently elaborated in the method description.

**Questions:**

1、Although the relationship between the anchoring model and the online model has been theoretically elucidated, their interactive mechanisms in practical applications may influence the final outcomes. Are there further details regarding how the anchoring model guides the attention mechanism of the online model? Does the anchoring model consistently demonstrate robust stability across diverse training scenarios?

2、This paper primarily validates the effectiveness of the ASMIL model in subtype classification and tumor localization tasks, but the multimodal learning framework can also demonstrate advantages in other tasks. Can this method be extended to other tasks?

3、The author proposes reducing overfitting risk by randomly discarding some feature labels, but this mechanism may impact model performance differently under various training settings. Could you provide more experimental data on how the label discard rate (e.g., the B value) affects model performance? Is there a recommended optimal discard rate, or should the discard proportion be adjusted based on different datasets?

---

> ### Author Response · Authors · 2025-11-23
>
> We thank the reviewer for the careful reading and positive evaluation of our work. Please see our detailed responses to your concerns as follows.
>
> **Weaknesses:**
>
> >1、The motivational section in Subsection 3.3 bears similarities to discussions on issues with attention mechanisms in related work. It requires further clarification of differences or highlighting of innovative aspects, and may also be subject to corresponding revisions or consolidation.
>
> We thank the reviewer for highlighting the overlap between Subsection 3.3 and the related work. In response, we have revised the related work section to focus on summarizing existing analyses of attention mechanisms and their limitations. Subsection 3.3 now serves exclusively to motivate ASMIL and to highlight its innovative aspects. Specifically, Subsection 3.3 contrasts our three identified issues—unstable attention dynamics, over-concentrated attention, and overfitting—with prior discussions, and emphasizes the novel aspects of our approach: the EMA anchor for stabilizing attention, the normalized sigmoid attention in the anchor model, and token-dropping regularization. We have removed redundant explanations and consolidated overlapping content between the two sections. These revisions clarify the distinction between prior work and our contributions, thereby eliminating redundancy and improving clarity.
>
> >2、The paper compares its results with multiple baseline models, but some of these comparison models (such as CLAM-SB and DSMIL) lack in-depth discussion regarding the rationale for their selection and their applicability. Key differences in design between these models and ASMIL, their respective strengths and weaknesses, and how these factors influence performance across different types of tasks remain unexplored.
>
> Thank you for this suggestion. In the revised manuscript, we have updated Section 2 Related Work to explicitly describe the rationale for selecting each baseline and to discuss its applicability to our problem setting. In particular, we now clarify that CLAM-SB and DSMIL are designed for weakly supervised multiple-instance-learning-based WSI subtyping, which aligns with our evaluation tasks. We further expand the comparison by outlining the key architectural differences between these methods and ASMIL—for example, CLAM-SB uses class-specific attention and instance pruning, DSMIL employs a dual-stream aggregation of instance- and bag-level predictions, whereas ASMIL introduces an EMA-updated anchor to stabilize attention distributions. We also discuss their respective strengths and limitations, supported by our experimental results, and explain how these design choices are reflected in performance across prediction and localization tasks.
>
> >3、The font in the model diagram (Figure 2) is partially obscured, and some words appear in a smaller font size that is not sufficiently clear.
>
> Thank you for pointing it out. In the revised manuscript, we have updated Figure 2 to address the obscured text and increased the label font size to ensure clear readability on screen and in print.

---

> ### Author Response · Authors · 2025-11-23
>
> **Weaknesses:**
> >4、The paper proposes updating the anchor model via EMA to stabilize the attention distribution of the online model, yet it does not delve deeply into the relationship between these two components. Specifically, the interaction mechanism between EMA updates and backpropagation, as well as the performance of the online model during inference, are not sufficiently elaborated in the method description.
>
> We discuss the structure of ASMIL in Section 4. The online model is updated by backpropagation, and the anchor model is updated using the EMA of the online model.
>
> To clarify the interaction, we have revised section 4.4 of the paper. Here, we also briefly describe the interaction between the optimizer step and the EMA update in Section 4. Using the same notation as the submission,  let $\theta_t$ denote the online model parameters and $\theta’_t$ the anchor model’s parameters at time step $t$. We first apply an optimizer step to update the online model:
>
> $$\theta_{t+1} = \theta_t -\eta \nabla_\theta \mathcal{L},$$
>
> where $\eta$ is the learning rate and $\mathcal{L}$ is the total loss. We then update the anchor  via an EMA of the updated online parameters:
>
> $$\theta_t’ \leftarrow m \theta_{t-1}’ + (1-m) \theta_t,$$
>
> where $m\in (0,1)$ is the EMA factor. No gradient flows through the anchor model.
>
> The anchor influences learning via the stability term in the loss function, which is the KL-divergence between the anchor’s and the online model’s attention distributions. As shown in Eq. 7 and Eq. 8, gradient descent on this KL term guides the online attention toward the anchor attention, so the EMA-smoothed anchor provides a stable reference that regularizes the online attention during training.
>
> At inference time, ASMIL uses only the online model, discarding the anchor model; thus, there is no interaction between them. We have revised Section 4 to clarify this point. We have already studied the impact of the EMA factor $m$ in Appendix K.3.
>
>
> **Questions:**
>
> >1、Although the relationship between the anchoring model and the online model has been theoretically elucidated, their interactive mechanisms in practical applications may influence the final outcomes. Are there further details regarding how the anchoring model guides the attention mechanism of the online model? Does the anchoring model consistently demonstrate robust stability across diverse training scenarios?
>
> Please refer to our response to Weakness 4.
>
> >2、This paper primarily validates the effectiveness of the ASMIL model in subtype classification and tumor localization tasks, but the multimodal learning framework can also demonstrate advantages in other tasks. Can this method be extended to other tasks?
>
> Thank you for your constructive question. The core ASMIL module is an attention-based MIL aggregator operating on bags of instance embeddings, and its anchor-based stabilization acts on the attention distribution itself. This design does not depend on the specific downstream task; only the final prediction head and loss function change.
> In the revision, we make this explicit in Section 4 by describing how ASMIL applies to both classification and survival prediction.
> Concretely, we add experiments on slide-level survival prediction across five public cohorts and report the full results in Appendix O. ASMIL achieves the best c-index across datasets, demonstrating that the proposed stability mechanism generalizes to a distinct task type beyond subtype classification and tumor localization.
> Regarding multimodal learning, we agree that combining WSI with additional modalities (e.g., clinical variables or genomic profiles) offers richer information for diagnosis and prognosis. In the current work, we deliberately restrict our experiments to WSI-based MIL to isolate and analyze the effect of attention stabilization. The design of multimodal encoders and fusion strategies is an orthogonal direction. We therefore revise the Future Work section to explicitly highlight multimodal ASMIL as a primary extension of this framework.
>
> >3、The author proposes reducing overfitting risk by randomly discarding some feature labels, but this mechanism may impact model performance differently under various training settings. Could you provide more experimental data on how the label discard rate (e.g., the B value) affects model performance? Is there a recommended optimal discard rate, or should the discard proportion be adjusted based on different datasets?
>
> We appreciate the reviewer’s attention to the effect of the label discard rate $B$. In our submission, **Appendix K.4** presents an ablation study on $B$ for both the BRACE dataset and CAMELYON-16. In most scenarios, validation performance is maximized at $B=0.5$. Consequently, we adopt $B=0.5$ as a robust default and apply it across all experiments in the main paper, without dataset-specific tuning. We have now clarified this design choice in Section 4 of the main text.

---

### Author Response · Authors · 2025-11-23

We sincerely thank all reviewers for their time, effort, and constructive feedback on our manuscript. In the revised version, we have carefully addressed each point raised in the reviews, and all corresponding changes in the manuscript are highlighted in blue for ease of reference.

---

### Meta-Review · Area_Chair_uCeB · 2026-01-03

**Summary:**

The paper was reviewed by 4 experts, with initial reviews as 6666.  The reviewers' concerns are listed in the next field, and all were addressed well by the reviewers. Most likely some of the reviewers would have increased their scores. The paper explores an important problem in ABMIL, which is the stability and robustness of the attention mechanism.

**Reviewer Concerns:**

**Reviewer JdSj**
1. motivation is similar to previous works, and could be consolidated better and innovative aspects better highlighted.
2. lacks in-depth discussion about the applicability of the compared models.
3. what is the interaction between EMA updates and back-prop? What its he performance of the online model during inference?
4. could it be applied to other tasks?
5. how does the discard rate affect the model training?

The AC thinks that all issues were addressed well.

**Reviewer e83y**
1. Why is the AUC 0.002 lower on Camelyon-17 w/ ResNet-18 backbone? Revise "SOTA on all datasets" claim?
2. Could NSF be applied to the online model using some heuristics, e.g., gradient clipping?
3. What is the sensitivity of m to the dataset characteristics?
4. does the optimal dropout rate hold for other WSI datasets? How does it affect the anchor model?
5. ablation studies comparing NSF with other normalization functions.
6. how to address the limitation of missing tiny tumors?

The AC thinks that all issues were addressed well. In addition, regarding Point 1, in Table 1, many differences between the top-two methods are small compared to the standard deviations, e.g., the 0.002 difference but the stddev is 0.061.  The AC highly encourages the authors to run statistical significance tests between the top-two methods in each column block to show that the differences are statistically significant or not.

**Reviewer Pd6q**
1. Both anchor and online model are intialized randomly. Why not initialize the anchor after a few epochs?
2. Could the dropout strategy be applied to other models? How does it compare to ICML2025 paper?

The AC thinks that all issues were addressed well.

**Reviewer YC75**
1. Better situate ASMIL within current teacher/EMA anchoring methods in self-supervised ViT literature.
2. Include correlation analysis to better show the stability-performance link.

The AC thinks that all issues were addressed well.

**Reviewer Scores:**

Most likely a majority of reviewers would have increased their scores.

---

### Decision · Program_Chairs · 2026-01-26

Accept (Poster)